# PPARγ mediated enhanced lipid biogenesis fuels *Mycobacterium tuberculosis* growth in a drug-tolerant hepatocyte environment

Binayak Sarkar[1], Jyotsna Singh[1], Mohit Yadav[1], Priya Sharma[2], Raman Deep Sharma[2], Shweta Singh[3], Aakash Chandramouli[4], Kritee Mehdiratta[5,6], Ashwani Kumar[3], Siddhesh S Kamat[4], Devram S Ghorpade[1], Debasisa Mohanty[1], Dhiraj Kumar[2], Rajesh S Gokhale[1,4*†]

[1]Immunometabolism Laboratory, National Institute of Immunology, New Delhi, India; [2]Cellular Immunology Group, International Centre for Genetic Engineering and Biotechnology, New Delhi, India; [3]CSIR-Institute of Microbial Technology, Chandigarh, India; [4]Indian Institute of Science Education and Research, Pune, India; [5]Council of Scientific and Industrial Research–Institute of Genomics and Integrative Biology (CSIR-IGIB), New Delhi, India; [6]Academy of Scientific and Innovative Research (AcSIR), Ghaziabad, India

*For correspondence:
rsg@nii.ac.in

Present address: †Indian Institute of Science Education and Research, Pune, India

## eLife Assessment

This **fundamental** study examines infection of the liver and hepatocytes during tuberculosis infection. The authors **convincingly** demonstrate that aerosol infection of mice and guinea pigs leads to appreciable infection of the liver as well as the lung. A further strength of the study lies in clinical evaluation of the presence of tuberculosis bacteria in human autopsied liver samples from individuals with miliary tuberculosis and the presence of a clear granuloma-like structure, which will prompt further study.

**Abstract** *Mycobacterium tuberculosis* (Mtb) infection of the lungs, besides producing prolonged cough with mucus, also causes progressive fatigue and cachexia with debilitating loss of muscle mass. While anti-tuberculosis (TB) drug therapy is directed toward eliminating bacilli, the treatment regimen ignores the systemic pathogenic derailments that probably dictate TB-associated mortality and morbidity. Presently, it is not understood whether Mtb spreads to metabolic organs and brings about these impairments. Here, we show that Mtb creates a replication-conducive milieu of lipid droplets in hepatocytes by upregulating transcription factor PPARγ and scavenging lipids from the host cells. In hepatocytes, Mtb shields itself against the common anti-TB drugs by inducing drug-metabolizing enzymes. Infection of the hepatocytes in the in vivo aerosol mice model can be consistently observed post-week, 4 along with enhanced expression of PPARγ and drug-metabolizing enzymes. Moreover, histopathological analysis indeed shows the presence of Mtb in hepatocytes along with granuloma-like structures in human biopsied liver sections. Hepatotropism of Mtb during the chronic infectious cycle results in immuno-metabolic dysregulation that could magnify local and systemic pathogenicity, altering clinical presentations.

## Introduction

*Mycobacterium tuberculosis* (Mtb), the causative agent of human tuberculosis, remains the leading infectious killer globally, with an estimated death of 1.3 million in 2022 (*World Health Organisation, 2023*). Despite progressive work on designing new anti-TB therapeutics and implementing vaccination programs in TB-endemic countries, it has a high global case fatality rate and a poor treatment success rate, along with a rising number of drug-resistant infections (*Seung et al., 2015*; *Husain et al., 2016*). Emerging paradigms in infectious diseases advocate tackling pathogen-driven ailments as one-dimensional problems, where the entire emphasis is given to pathogen elimination. A holistic understanding of how the host systems respond to the infection, vaccination, and treatment is key to TB management programs (*Eckhardt et al., 2020*). Recent widespread and severe physiological derangements associated with COVID-19 patients, even after the elimination of the virus, have brought back the focus on identifying novel strategies that are inclusive of modulating the host immune-metabolic axis (*Davis et al., 2023*; *Lippi et al., 2023*).

The clinical symptoms of pulmonary TB encompass localized manifestations like prolonged cough with mucus, pleuritic chest pain, hemoptysis, and lung damage. Besides, systemic outcomes like cachexia, progressive fatigue, oxidative stress, altered microbiota, and glucose intolerance result in organ-wide disruptions (*Luies and du Preez, 2020*; *Loddenkemper et al., 2015*). Pulmonary TB patients often suffer from progressive and debilitating loss of muscle mass and function, with severe weight loss, this TB-associated cachexia cannot be reversed by conventional nutritional support (*Morley et al., 2006*; *Chang et al., 2013*). Besides, numerous epidemiological studies indicate that hyperglycemia may occur during active tuberculosis, which can compromise insulin resistance and glucose tolerance, although the mechanisms are unclear (*Luies and du Preez, 2020*; *Bisht et al., 2023*; *Niazi and Kalra, 2012*). Both the localized and systemic pathophysiology of TB infection indicate an alteration in the host immuno-metabolic axis. It is somewhat bewildering that the engagement of the liver during the Mtb infection cycle is not considered, despite its central role in balancing the immune and metabolic functions of the body (*Jensen-Cody and Potthoff, 2021*). The crosstalk between the liver and lung has been largely overlooked in TB, even though acute phase proteins (APPs) are used as predictive biomarkers in pulmonary tuberculosis (*Kumar et al., 2021*). A robust hepatic APR response in mice, mediated by key hepatocyte transcription factors, STAT3 and NF-κB/RelA, has been known to trigger pulmonary host defences for survival during pneumonia and sepsis (*Quinton et al., 2018*). In TB, the active phase of the disease is associated with heightened expression of various genes that modulate flux in the lipid metabolic pathways (*Shim et al., 2020*; *Park et al., 2021*; *Gago et al., 2018*).

The liver is involved in a wide variety of functions – synthesis of plasma proteins, secretion of various hepatokines, degradation of xenobiotic compounds, and storage of lipids, glucose, vitamins, and minerals (*Wang et al., 2021*; *Alper et al., 1969*). De novo lipogenesis, secretion of acute phase proteins, hepatokine production, etc., are all directly or indirectly controlled by the hepatocytes, thereby communicating with almost all the organs of the body (*Wang et al., 2021*; *Zhou et al., 2016*). Moreover, the liver is actively involved in triacylglycerol synthesis and storage under the intricate regulation of various hormones like insulin, glucagon, and thyroid hormone (*Zhang et al., 2022*). All these functions uniquely position the liver as the central regulator of lipid metabolism. To avert organ damage, the liver maintains tolerogenic properties rendering it an attractive target for various pathogenic microorganisms. Although several studies have indicated the role of both -Mtb virulence components and host factors (immune activation, nitric oxide, IFNγ, intracellular pH, and hypoxia) in the generation of Mtb drug tolerance, the role of liver, being the principal center for xenobiotic metabolism needs careful investigation (*Datta et al., 2024*; *Deb et al., 2009*; *Liu et al., 2016*; *Samuels et al., 2022*; *Santucci et al., 2021*). Liver is the hub of both phase I and phase II drug-modifying enzymes (DMEs). The levels as well as the activity of both types of DMEs play significant roles in determining the pharmacokinetics and efficacy of various drugs across multiple diseases, from infection to malignancy (*Wu and Lin, 2019*).

In this study, we demonstrate the active involvement of the liver in a murine aerosol TB infection model during the chronic phase and establish hepatocytes as a new replicative niche for Mtb. Using a variety of in vivo, ex vivo, and in vitro techniques, we show how Mtb perturbs biological functions within hepatocytes remodeling intracellular growth, localization, and drug sensitivity. Cellular and mass spectrometric studies demonstrate Mtb infection-mediated enhanced fatty acid biogenesis and

TAG biosynthesis in the hepatocytes regulated by PPARγ. We propose that infection of hepatocytes by Mtb during the chronic phase can contribute to significant changes in disease progression, TB treatment, and the development of infection-induced metabolic diseases.

## Results

### Human miliary tuberculosis patients harbor Mtb in the liver

Mtb infects lungs, and other organs like lymph nodes, pleura, bones, and meninges. There are also isolated case reports of hepatic TB, without providing many pathophysiological consequences (*Wu et al., 2013*; *Turhan et al., 2011*). To gain further insights into the involvement of the liver in Mtb infections, we acquired human autopsied liver samples from individuals with miliary tuberculosis and analyzed them for the presence of Mtb bacilli (the details of the human samples have been included in *Supplementary file 1*). Hematoxylin and eosin (H and E) staining showed the presence of distinct immune cell infiltration and granuloma-like structures in the infected samples (*Figure 1A* and *Figure 1—figure supplement 1C*). We examined the liver specimens, with Mtb-specific Ziehl-Nielsen (Z-N) acid-fast and auramine O-rhodamine B stain (*Figure 1B and C*). Both these stains showed distinct positive signals with characteristic rod-shaped bacilli that could be visualized by the acid-fast staining (as indicated by arrows) (*Figure 1B*). We further corroborated our findings by performing fluorescence in situ hybridization (FISH) using Mtb-specific 16 s rRNA probes, where specific signals were observed (*Figure 1D*). The specific staining in FISH eliminates the possibility of non-tuberculous mycobacteria (NTMs) in the tissue specimen and confirms the presence of Mtb infection in the liver. Similar staining in the uninfected liver sections from other individuals did not show any signal (*Figure 1—figure supplement 1A and B*). Although hepatic granuloma is the characteristic histological feature for both local and miliary forms of hepatic TB, the precise involvement of the different cells like Kupffer cells, hepatocytes, stellate cells, liver sinusoidal endothelial cells, hepatic stellate cells, and other cell types has not been studied in detail (*Hickey et al., 2015*). Multiplex immunostaining with β-actin antibody and Mtb-specific Ag85B antibody shows the presence of Ag85B signals within the human hepatocytes, further confirming the presence of Mtb within hepatocytes (*Figure 1E* and *Figure 1—figure supplement 1D*), indicated with yellow arrows. Hepatocytes are morphologically distinct, large polygonal cells (20–30 µm) with round nuclei, many of which are double-nucleated and mainly positioned at the center of the cytoplasm (*Celton-Morizur et al., 2010*). The presence of Ag85B signals near the nucleus, as depicted in *Figure 1E* and *Figure 1—figure supplement 1D*, further supports our assertion. Moreover, hepatic granulomas in the human samples showed localized clustering of the immune cells (*Figure 1—figure supplement 1D*). The corresponding Ag85B staining in the uninfected liver biopsy samples did not show any signal (*Figure 1F*). Furthermore, to establish a correlation between liver Mtb load and Mtb burden at the primary infection site, the lung, we conducted H&E staining and acid-fast staining on lung sections from the same individuals. H&E staining revealed distinct granulomas, while acid-fast staining confirmed an elevated bacterial load, both indicative of a high degree of infection (*Figure 1—figure supplement 1E and F*). Using auramine O-rhodamine B, Ziehl-Nielsen acid-fast staining, and FISH, Mtb presence was detected in several lung specimens (*Figure 1—figure supplement 1G*). These results indicate Mtb infection in the liver of human subjects and suggest the localization of Mtb within hepatocytes. These findings are quite intriguing considering very few studies have discussed the lung-liver crosstalk or the involvement of liver in pulmonary TB.

### Mtb infection of mice via the aerosol route leads to significant infection of the liver and primary hepatocytes

To investigate whether the liver harbours Mtb during mice aerosol infection, we infected C57BL/6 mice with 200 CFU of Mtb *H37Rv* and scored for the bacterial load in the conventional niche- the lung as well as in the liver. Mtb could be detected in the liver consistently across several experiments at 4 weeks post-infection and the bacterial load increased till week 10 (*Figure 2A and B*). Consistent with the Mtb burden, phalloidin and hematoxylin and eosin (H and E) staining of the infected liver at 8 weeks post-infection showed localized cellular aggregation forming ectopic granulomas forming a granuloma-like structure (*Figure 2C* and *Figure 2—figure supplement 1C*). Further staining the liver sections with CD45.2 antibody shows clustering of immune cells in Mtb-infected mice liver (*Figure 2—figure supplement 1D*). To assess whether liver infection leads to deranged liver function in the

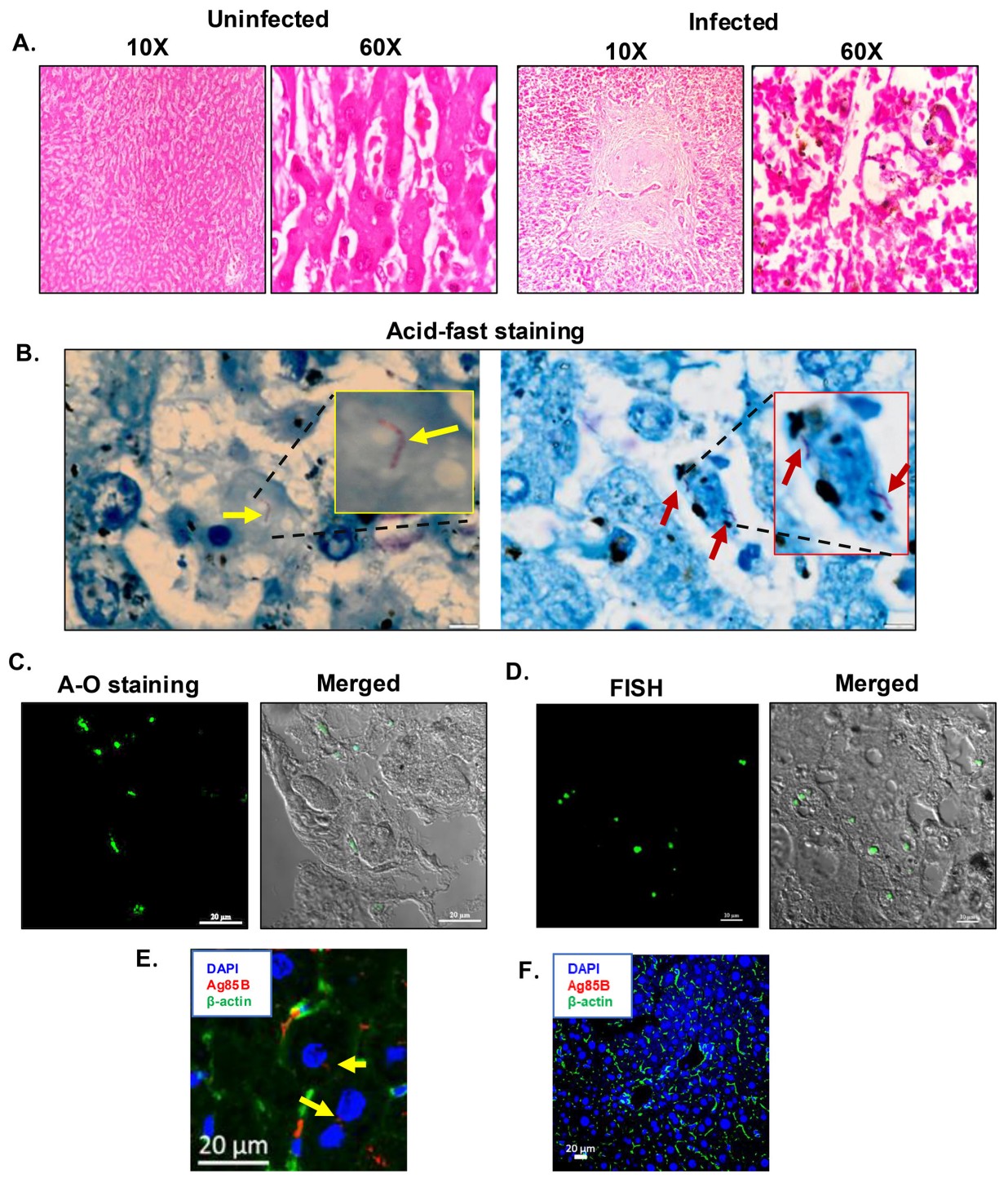

**Figure 1.** Presence of *Mycobacterium tuberculosis* (Mtb) in the liver of human miliary tuberculosis patients. (**A**) Hematoxylin and eosin (H and E) staining of the human autopsied liver tissue sections showing the presence of granuloma-like structure in the Mtb-infected liver. (**B**) Acid-fast staining shows the presence of Mtb in the liver section of miliary tuberculosis (TB) patients (arrows point to the presence of Mtb in the enlarged image). (**C–D**) Auramine O-Rhodamine B (A–O) staining and fluorescence in situ hybridization (FISH) with Mtb-specific 16 s rRNA primer further confirms the presence of Mtb in human liver tissue sections. (**E**) Dual staining of β-actin (green) and Ag85B (red) using respective antibodies shows the presence of Mtb in hepatocytes of human biopsied liver sections. (**F**) Dual staining of β-actin (green) and Ag85B (red) using respective antibodies shows no distinct signal of Ag85B in the uninfected liver biopsied sample (control). The data is representative from six human patient samples.

The online version of this article includes the following figure supplement(s) for figure 1:

**Figure supplement 1.** Histopathological changes in the liver biopsy samples of miliary tuberculosis (TB) patients: (**A and B**) A-O staining and fluorescence in situ hybridization (FISH) in the liver biopsy of the uninfected (control) group shows no *Mycobacterium tuberculosis* (Mtb)-specific signals.

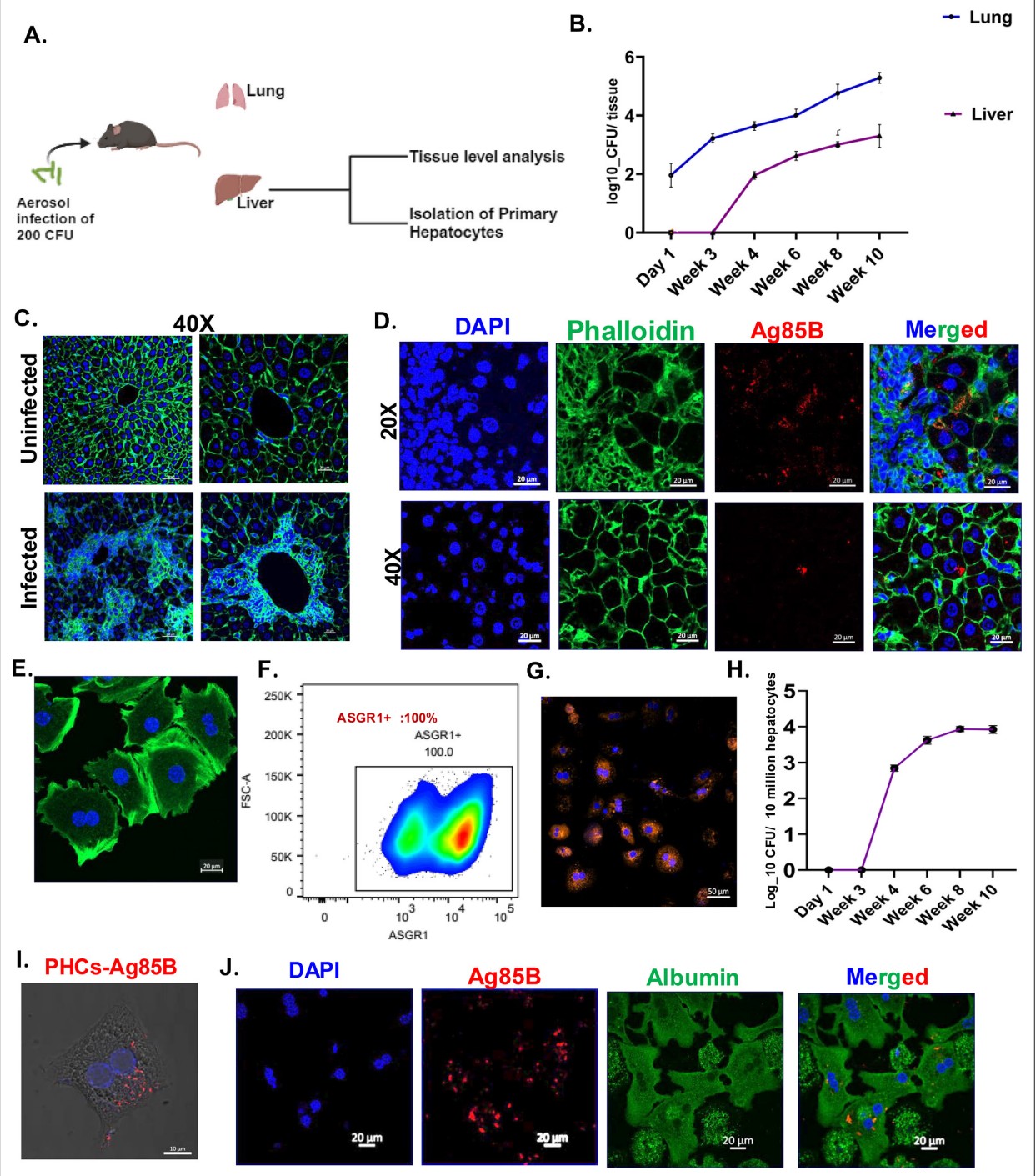

**Figure 2.** Involvement of the liver and hepatocytes in a mouse aerosol model of *Mycobacterium tuberculosis* (Mtb) infection. (**A**) Schematic denoting the flow of experimental set up of studying the liver at tissue level and cellular level (generated using Biorender.com). (**B**) C57BL/6 mice were infected with 200 colony-forming units (CFU) of H37Rv through the aerosol route and the bacterial burden of the lung and liver was enumerated at different time points post-infection in lung and spleen. n=5 mice/group and the data is representative from three independent biological experiments (**C**) Alexa Fluor 488 Phalloidin and DAPI staining of uninfected and infected lungs at 8 weeks post-infection. Images were taken in 40 X magnification as mentioned (scale bar is 20 μm). (**D**) Immunofluorescence staining of Alexa Fluor 488 Phalloidin, DAPI, and Ag85B in the infected mice liver at 8 weeks post-infection shows the presence of Ag85B signals within hepatocytes (magnified image) (scale bar is 20 μm). (**E**) Isolated primary hepatocytes stained with phalloidin green and DAPI show the typical polygonal shape with binucleated morphology. (**F**) Anti-asialoglycoprotein receptor (ASGR1) antibody staining specifically stains primary hepatocytes isolated from infected mice, as validated by the contour plot in flow cytometry. (**G**) Antiasialoglycoprotein receptor (ASGR1) antibody staining specifically stains primary hepatocytes isolated from mice, as visualized through confocal microscopy. (**H**) Primary

*Figure 2 continued on next page*

*Figure 2 continued*

hepatocytes were isolated from the infected mice, lysed and CFU enumeration was done at the mentioned time points. (**I**) Ag85B staining of cultured primary hepatocytes, isolated from mice at 8 weeks post-Mtb infection. (Scale bar is 20 μm). (**J**) Dual immunostaining of Ag85B (red) and albumin (green) shows distinct signals of Mtb within the isolated hepatocytes from infected mice at 8 weeks post-infection. The figures show representative data from four independent biological experiments.

The online version of this article includes the following source data and figure supplement(s) for figure 2:

**Source data 1.** Data used for plotting the graph in *Figure 2B*.

**Source data 2.** Data used for plotting the graph in *Figure 2H*.

**Figure supplement 1.** Histopathological changes in the murine liver post *Mycobacterium tuberculosis* (Mtb) infection.

**Figure supplement 1—source data 1.** Data used for generating the graph in *Figure 2—figure supplement 1A*.

**Figure supplement 1—source data 2.** Data used for generating the graph in *Figure 2—figure supplement 1B*.

**Figure supplement 1—source data 3.** Data used for generating the graph in *Figure 2—figure supplement 1E*.

**Figure supplement 2.** Involvement of the liver in the guinea pig aerosol infection model.

**Figure supplement 2—source data 1.** Data used for generating the plot in *Figure 2—figure supplement 2A*.

aerosol model, we analyzed the levels of liver functional enzymes like albumin, aspartate transaminase (AST), alanine transaminase (ALT), and gamma-glutamyl transpeptidase (GGT) in the sera. Till week 10 post-infection, sera levels of these markers showed significant alterations as the infection progressed, especially in the levels of AST with changes also in the levels of albumin and GGT (*Figure 2—figure supplement 1B*). Since hepatocytes, the principal parenchymal cells of the liver constitute 70–80 % of the liver by weight, we hypothesized whether hepatocytes could harbour Mtb in the murine model of infection. To this end, we isolated primary hepatocytes from the Mtb-infected mice at different time points post-infection. The purity of the isolated hepatocytes was validated by multiple methods-morphologically hepatocytes can be identified by their distinct hexagonal architecture, round nucleus, some of which are binucleated, as observed in Alexa fluor phalloidin 488 and DAPI-stained hepatocytes (*Figure 2E*). Flow cytometry staining with antibody against hepatocyte-specific marker asialoglycoprotein receptor 1 (ASGR1) protein, confirmed that almost 100 % of the isolated cells are primary hepatocytes as seen in (*Figure 2F*). It was further validated by confocal microscopy as all the cultured cells specifically stained for ASGR1 protein, although with variable levels of expression (*Figure 2G*). After thorough confirmation of the identity and the purity of the hepatocytes, the cells were lysed and plated. Like the whole liver CFU, Mtb-infected primary hepatocytes (PHCs) at 4 weeks post-infection with substantial bacterial load at week 6 and week 8 (*Figure 2H*). Staining with antibody for Mtb-specific Ag85B protein in both the infected tissue sections and cultured hepatocytes isolated from the in vivo infected mice revealed the presence of Mtb within hepatocytes (*Figure 2D and I*). To further strengthen our data, we co-stained the isolated primary hepatocytes from the Mtb-infected mice with anti-albumin antibody (albumin is a robust and widely used hepatocyte marker) and Ag85B. As depicted in (*Figure 2J*)**,** distinct Ag85B signals can be observed in the albumin-rich hepatocytes. Even at a low CFU of 50, Mtb load was observed in the liver at week 6 and 8 (*Figure 2—figure supplement 1E*). Besides the aerosol route, infection intraperitoneally also led to hepatocyte infection, also led to hepatocyte infection as early as day 10 (*Figure 2—figure supplement 1A*). Further, to prove the spread of Mtb to the liver in other model organisms, we infected guinea pigs with 200 CFU of Mtb and analyzed the bacterial load in the lung, liver, and spleen at week 4 and week 8 post-infection (*Figure 2—figure supplement 2A*). At both the time points, we could observe robust liver infection with granulomatous structure in the liver, proving that the dissemination of Mtb to the liver occurs across multiple model animals (*Figure 2—figure supplement 2B*).

## Hepatocytes provide a replicative niche to Mtb

Consistent data from human TB patients and aerosol murine model prompted us to develop an in vitro model of Mtb and hepatocyte infection. We examined Mtb infection in several hepatocyte cell lines of mouse and human origin, along with primary murine hepatocytes. Infection studies with fluorescently labeled Mtb *H37Rv* were carried out with primary murine hepatocyte cells (PHCs), HepG2, Huh-7, and AML-12 (*Figure 3A, B and C*). The multiplicity of infection was titrated both in PHCs and HepG2 with the MOI of 1, 2.5, 5, and 10. MOI 10 consistently showed an infectivity of more than 60% in both

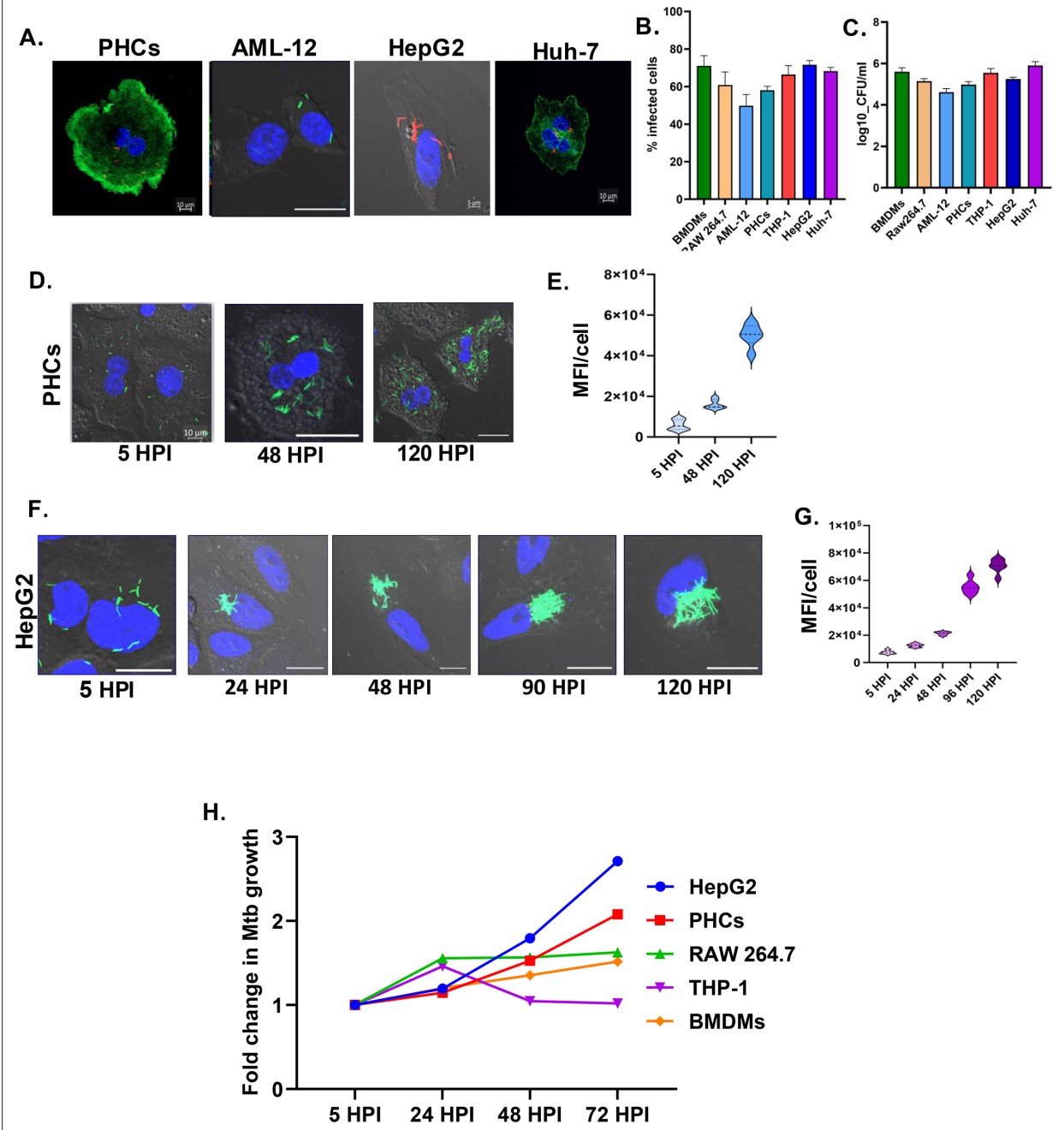

**Figure 3.** *Mycobacterium tuberculosis* (Mtb) uses hepatocytes as a replicative niche. (**A and B**) Representative microscopic images showing the infection of primary hepatocytes (PHCs), AML-12, HepG2, and Huh-7 with labeled Mtb-H37Rv strains and subsequent quantification of percentage infectivity in the respective cell types. RAW 264.7, THP-1, and murine bone marrow-derived macrophages (BMDMs) were used as macrophage controls. The scale bar in all images is 10 µm except in HepG2, which is 5 µm (**C**) colony-forming unit (CFU) enumeration of Mtb-H37Rv in different hepatic cell lines. (**D and E**) Representative confocal microscopy images of Mtb-GFP-H37Rv infected PHCs at the respective time points post-infection and bar blot depicting mean fluorescent intensity (MFI)/cell at the respective time points. (**F and G**) Representative confocal microscopy images of MtbH37Rv-GFP infected HepG2 at the respective time points post-infection and bar blot depicting MFI/cell at the respective time points, n=4 independent experiments, with each dot representing five fields of 30–60 cells. Scale bar 10 µm. (**H**) Fold change of growth rate of Mtb within HepG2, PHCs, RAW 264.7, THP-1, and BMDMs at the mentioned time points post-infection. All the figures show representative data from four independent biological experiments.

The online version of this article includes the following source data and figure supplement(s) for figure 3:

**Source data 1.** Data used for generating the graph in *Figure 3B*.

*Figure 3 continued*

**Source data 2.** Data used for generating the graph in *Figure 3C*.

**Source data 3.** Data used for generating the graph in *Figure 3E*.

**Source data 4.** Data used for generating the graph in *Figure 3G*.

**Source data 5.** Data used for generating the graph in *Figure 3H*.

**Figure supplement 1.** Standardization of multiplicity of infection (MOI) in primary hepatocytes (PHCs) and HepG2.

**Figure supplement 1—source data 1.** Data used for generating *Figure 3—figure supplement 1B*.

**Figure supplement 1—source data 2.** Data used for generating *Figure 3—figure supplement 1C*.

**Figure supplement 1—source data 3.** Data used for generating *Figure 3—figure supplement 1D*.

**Figure supplement 1—source data 4.** Data used for generating *Figure 3—figure supplement 1E*.

**Figure supplement 1—source data 5.** Data used for generating *Figure 3—figure supplement 1F*.

**Figure supplement 1—source data 6.** Data used for generating *Figure 3—figure supplement 1G*.

**Figure supplement 1—source data 7.** Data used for generating *Figure 3—figure supplement 1H*.

HepG2 and PHCs (*Figure 3—figure supplement 1A, B and C*). While we acknowledge that MOI 10 is moderately high, several well-cited studies have used MOI 10 for various conventional phagocytes and other non-conventional cell types like mesenchymal stem cells, adipocytes, and human lymphatic endothelial cells (*Jain et al., 2020*; *Lerner et al., 2020*; *Kalam et al., 2017*; *Peyron et al., 2008*; *Beigier-Bompadre et al., 2017*). Hence, MOI 10 was selected for further experiments. Macrophage cell lines RAW 264.7, THP-1, and mouse bone marrow-derived macrophages (BMDMs) were used as positive controls. Even though PHCs, HepG2, Huh-7, and AML-12 are not considered to be classical phagocytic cells, all cells showed infectivity of more than 60% after 24 hr, comparable to RAW 264.7, THP-1, and BMDMs (*Figure 3A and B*). Analysis of bacterial load in the PHCs post-infection showed colony-forming units (CFU) like RAW 264.7, THP-1, and BMDMs, supporting microscopic observations (*Figure 3C*). Mtb in macrophages is known to remodel the intracellular environment to survive within phagosomes. We studied Mtb growth kinetics within hepatocytes using GFP-labeled Mtb *H37Rv* in PHCs and HepG2 (*Figure 3D and F*). Mean fluorescent intensity measurements showed a consistent increase in GFP intensity in both PHCs and HepG2 with increasing time (*Figure 3E and G*). Fold change in replication dynamics with respect to 5 hr post-infection (HPI) showed that, while bacterial growth in macrophages plateaus after 48 hr post-infection, Mtb continues to grow in both HepG2 and PHCs (*Figure 3H*). Relative CFU numbers in the PHCs and HepG2 further supports our data (*Figure 3—figure supplement 1D-H*). Our studies thus establish that hepatocytes, besides being robustly infected by Mtb, also provide a favourable replicative niche for Mtb.

## Transcriptomics of infected hepatocytes reveal significant changes in key metabolic pathways

To understand Mtb-induced changes in the hepatocytes and the underlying mechanisms of how hepatocytes provide a favourable environment to the pathogen, we performed transcriptomic analysis of the infected and sorted HepG2 cells at 0 hr (5 hr post-incubation) and 48 hr post-infection. Sorting before RNA isolation specifically enriches the infected cellular population, thus eliminating cellular RNA from uninfected cells (*Figure 4A*). Unsupervised clustering segregated the data into 4 distinct groups on the PC1 with a variance of 27 %, showing good concordance within the replicates (*Figure 4B*). The close spatial clustering for the two 0 hr time points corresponding to uninfected and infected is indicative of relatively less transcriptomic changes. On the other hand, the spatial segregation of the 48 hr datasets suggests clear differences between the RNA transcripts of uninfected and infected cells. The differentially expressed genes were calculated using DESeqR with fold change >0.5 and a false discovery rate of <0.2. Gene ontology (GO) enrichment analysis for the differentially regulated pathways at both the early (0 hr) and the late (48 hr) infection time points is shown in (*Figure 4C*). At 0 hr post-infection, the immediate stress response pathway of the cell, involving ROS generation, intracellular receptor signaling pathways, and response to xenobiotic stresses got activated, while at a late time point, Mtb modulated some of the key immuno-metabolic pathways like macroautophagy, cellular respiration, proteasomal degradation pathway, response to type I interferon, IκB kinase/NF-κB signaling, etc (*Figure 4D*). Major alterations in the vacuolar and vesicular

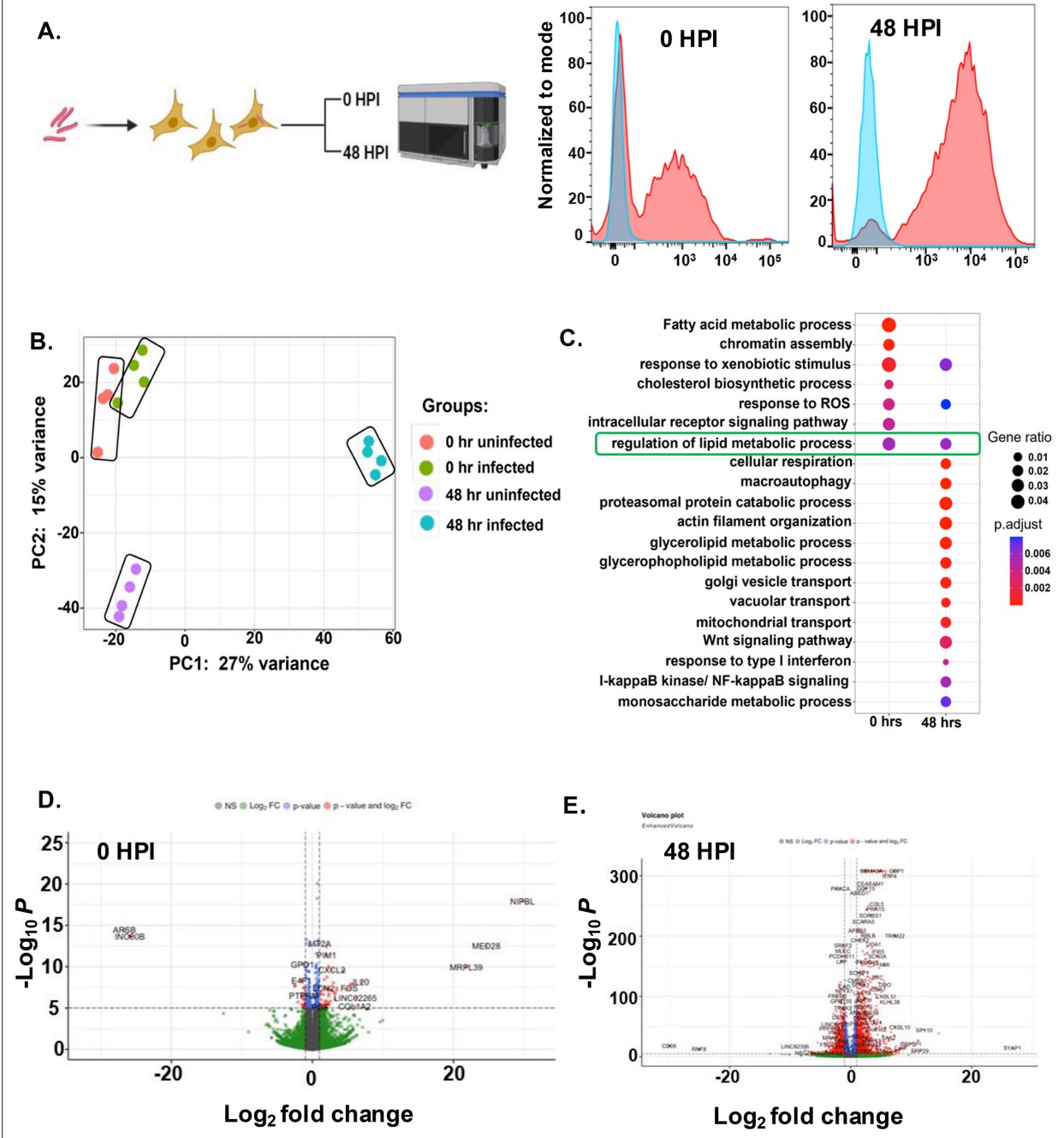

**Figure 4.** RNA sequence analysis of infected and sorted hepatocytes at 0 hr and 48 hr post-infection. (**A**) Experimental setup for infection of HepG2 with Mb-H37Rv-mCherry at 10 multiplicity of infection (MOI) with histogram of mCherry signals at 0- and 48 hr post-infection (schematic depicting experimental setup generated using Biorender.com) (**B**) Principal component analysis (PCA) plot illustrating the HepG2 transcriptome, identified through global transcriptomic analysis of 0 hr uninfected and infected and 48 hr uninfected and infected. (**C**) Gene ontology (GO) pathway enrichment analysis was done for differentially expressed genes (DEGs) with adjusted *o*-value <0.05 at 0- and 48 hr post-infection. The bubble plot depicts the enrichment of pathways on the mentioned time points post-infection, where the coordinate on the x-axis represents the gene ratio, the bubble size represents the gene count, and color represents the p-value. (**D and E**) Volcano plot depicting the fold change of different genes in 0 hr and 48 hr post-infection, the red dots represent the significantly upregulated and downregulated genes.

The online version of this article includes the following source data and figure supplement(s) for figure 4:

**Source data 1.** Data used to generate the volcano plot in *Figure 4D*.

**Source data 2.** Data used to generate the volcano plot in *Figure 4E*.

**Figure supplement 1.** Comparison of significant pathways between *Mycobacterium tuberculosis* (Mtb)-infected THP-1 and HepG2, 48 HPI.

transport at 48 hr are indicative of the dynamic changes in the phagosome maturation pathway (*Figure 4D*). Volcano plot analysis showed greater relative changes in the gene expression pattern at 48 hr compared to 0 hr, with many genes like *CXCL10, CXCL11, IDO, CCL5*, etc being greatly upregulated (*Figure 4E and F*). Interestingly, our RNA sequencing data indicated major changes in various facets of lipid metabolic pathways like fatty acid biosynthesis pathway, glycerolipid and glycerophospholipid metabolism, cholesterol biosynthesis pathways, etc. Several key genes like *FASN, DGAT1, DGAT2, HMGCR*, etc., were upregulated, indicating the possibility of greater synthesis of neutral lipids. Additionally, we have added a supplementary excel file with top 1000 differentially expressed genes (DEGs) at both 0 hr and 48 hr post-infection (*Supplementary file 2*). Thus, transcriptomic studies shed light on several of the key Mtb-induced changes in the hepatocytes.

To understand the difference and similarities between the pathways that get affected in macrophages compared to hepatocytes during Mtb infection, we compared the gene expression analysis data from Mtb-infected HepG2, 48 hr post-infection with THP-1 infected macrophages, 48 post-infection, taken from *Kalam et al., 2017*; *Figure 4—figure supplement 1A and B*. The comparison provides insights regarding how Mtb modulates different pathways depending on the type of the host cell infected. In THP-1, most of the altered pathways are related to mounting an effective immune response to the bacteria like type 1 interferon signalling pathway, regulation of cytokine secretion, leukocyte chemotaxis, response to zinc, etc, while in HepG2 the pathways that are getting altered are related to metabolism like response to xenobiotic stimulus, macroautophagy, glycerophospholipid and glycerolipid metabolism, and cellular respiration. This comparative analysis is indicative of Mtb's ability to harness the metabolic richness of hepatocytes, probably as a source for nutrients. Moreover, being a non-immune cell type, hepatocytes might lack a robust innate immune pathway like the macrophages and hence be less likely to clear mycobacterial infections.

## Increased fatty acid synthesis drives Mtb growth in hepatocytes

Mtb survival in foamy macrophages is driven by nutrient acquisition from the lipid droplets (*Peyron et al., 2008*). Transcriptomic studies of Mtb-infected cells also showed upregulated pathways for lipid metabolism. Examination of Lipid droplets in both PHCs and HepG2 revealed an increase in the number of lipid droplets at 24 hr post-infection (*Figure 5A and B*). Time kinetic analysis of BODIPY intensity in the infected HepG2 at different days post-infection indicated a concomitant increase in lipid droplets with the progress of infection (*Figure 5C*). Moreover, in PHCs, GFP-labeled Mtb showed a high degree of colocalization with the lipid droplets (*Figure 5D*). Lipid droplets are single membrane-bound depots consisting mainly of neutral lipids like diacylglycerols (DAGs), triacylglycerols (TAGs), and cholesterol esters (CEs) (*Wölk and Fedorova, 2024*).

Mass-spectrometric analysis of the infected and uninfected HepG2 cells at 24 hr post-infection, showed an increase in both TAGs and DAGs and CEs with a decrease in the levels of free cholesterols, indicating Mtb-induced changes in the neutral lipid biosynthesis (*Figure 5E*). To understand whether Mtb utilizes host lipid droplets as a source of nutrients in hepatocytes, we treated Mtb-infected hepatocytes with specific inhibitors of de novo fatty acid biosynthesis (C75) and TAG biosynthesis (T863) (*Figure 5—figure supplement 1A*). Interestingly, inhibiting both de novo fatty acid biosynthesis as well as TAG biosynthesis reduced the bacterial load in both PHCs and HepG2 by almost 1.0–1.5 log fold (*Figure 5G and H*). In THP-1 macrophage, although C75 reduced bacterial load by 0.5 log fold but T863 did not affect the bacterial load (*Figure 5I*). To get better insights into the nutritional dependency of intracellular Mtb on hepatocyte lipid source, we metabolically labeled HepG2 with 7.5 µg/ml of fluorescently tagged fatty acid (BODIPY 558/568 $C_{12}$) which subsequently accumulated into host lipid droplets. After 16–20 hr, the labeled cells were treated with T863 and C75. Although T863 significantly reduced the load of the intracellular TAGs, C75 had little effect on the level of accumulated TAGs and showed a phenotype like the DMSO control (*Figure 5—figure supplement 1C and D*). We infected these three sets of cells with Mtb (MOI:10) for 24 hr. The Mtb isolated from DMSO and C75-treated HepG2 became distinctly labeled with highly fluorescent lipid bodies, but the fluorescence signal in Mtb derived from T863-treated cells was quite low and sparse (*Figure 5F*, *Figure 5—figure supplement 2A*). Considering that host-derived fatty acids are converted to TAGs in the bacterial cytoplasm as a source of carbon and energy, we wanted to check whether the TAG synthesis machinery was modulated in the hepatocyte-infected Mtb. To this end, we isolated Mtb from HepG2, 48 hr post-infection, and analysed for key genes involved in TAG biosynthesis. Mtb

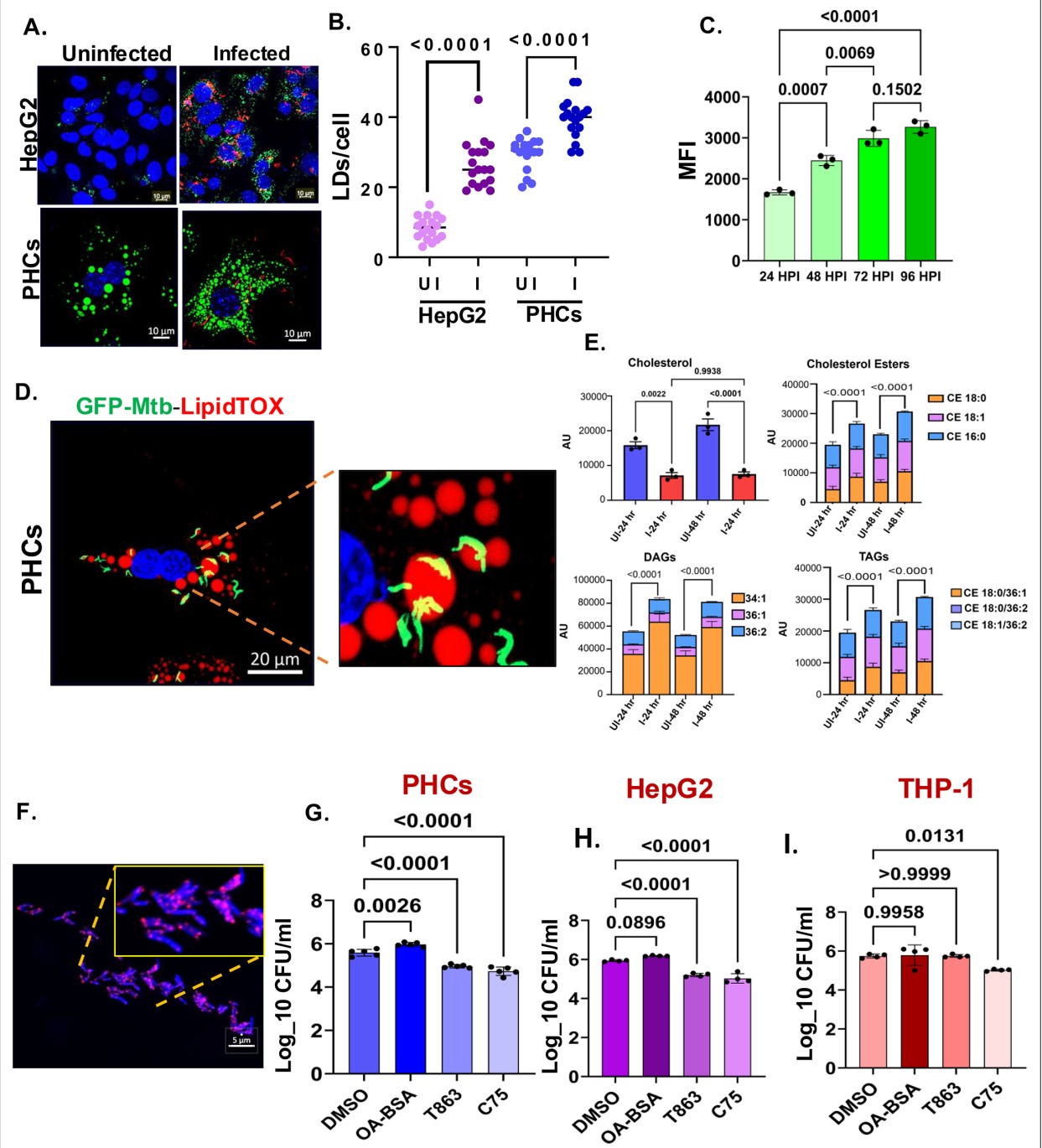

**Figure 5.** Increased fatty acid biogenesis drives *Mycobacterium tuberculosis* (Mtb) growth in hepatocytes. (**A**) Increased number of LDs/cells in HepG2 and primary hepatocytes (PHCs) post-Mtb infection as compared to the uninfected cells (**B**) Quantification of the number of lipid droplets in the infected HepG2 and PHCs with their uninfected control, 50–70 cells were analyzed from 3 independent experiments. (**C**) Increase in the BODIPY intensity at different days post-infection in infected hepatocytes (**D**). A high degree of colocalization of Mtb-*H37Rv*-GFP (green) with lipid droplets (red) within PHCs (**E**) Mass spectrometric quantification of the different species of diacylglycerols (DAGs), triacylglycerols (TAGs), and cholesterol esters in the Mtb-infected hepatocytes. (**F**) Confocal images showing puncta of fluorescently labeled fatty acid in Mtb derived from metabolically labeled (with BODIPY 558/568 C$_{12}$) HepG2. Relative colony-forming unit (CFU) of Mtb under the administration of different inhibitors of the lipid metabolic pathway in (**G**). PHCs (**H**). HepG2 and (**I**). THP-1. Representative data from three independent biological experiments. Data were analyzed using the two-tailed unpaired Student's t-test in B and one-way ANOVA in C, F, G, and H.*$p<0.05$, **$p<0.005$, ***$p<0.0005$, ****$p<0.0001$. ns = non-significant. The figures represent data from three independent biological experiments.

The online version of this article includes the following source data and figure supplement(s) for figure 5:

*Figure 5 continued on next page*

*Figure 5 continued*

**Source data 1.** Data used for generating the graph in *Figure 5B*.

**Source data 2.** Data used for generating the graph in *Figure 5C*.

**Source data 3.** Data used for generating the graphs in *Figure 5E*.

**Source data 4.** Data used for generating the graph in *Figure 5G*.

**Source data 5.** Data used for generating the graph in *Figure 5H*.

**Source data 6.** Data used for generating the graph in *Figure 5I*.

**Figure supplement 1.** Increased lipid biogenesis provides a growth advantage to *Mycobacterium tuberculosis* (Mtb) within hepatocytes.

**Figure supplement 1—source data 1.** Data used for generating *Figure 5—figure supplement 1B*.

**Figure supplement 2.** Confocal microscopy images showing the puncta of fluorescently labeled fatty acid (BODIPY 558/568 C12) in *Mycobacterium tuberculosis* (Mtb) isolated from metabolically labeled HepG2.

**Figure supplement 3.** Enhanced lipid accumulation in the liver of *Mycobacterium tuberculosis* (Mtb)-infected mice.

**Figure supplement 3—source data 1.** Data used for generating the plot *Figure 5—figure supplement 3D*.

grown in DMEM was used as a control. Interestingly, *Tgs1, Tgs4, Rv 1760*, etc, were upregulated by 6–8-fold, indicating an active Mtb transcriptional change in Mtb to utilize and store host-derived lipids (*Figure 5—figure supplement 1B*). Cumulatively, our studies thus demonstrate fatty acid biosynthesis and TAG formation to be important for Mtb growth in hepatocytes.

To assess the status of the neutral lipids in the liver, BODIPY staining was conducted in the liver cryosections of uninfected and infected mice (8 weeks post-infection). The liver of Mtb-infected mice bears more lipid bodies (*Figure 5—figure supplement 3A*). Co-immunostaining of BODIPY and Ag85B also showed abundant lipid droplets and specific signals of Ag85B (*Figure 5—figure supplement 3B*). Next, we stained the liver of Mtb-infected mice at 2-, 4-, and 8 weeks post-infection with LipidTOX neutral red dye. Surprisingly we found an elevated signal intensity of the dye at 8 weeks post-infection. This shows that in mice, lipid droplets might correlate with Mtb burden (*Figure 5—figure supplement 3C, D*). Our data comprehensively establishes infection-induced lipid droplet accumulation as a pathogenic outcome of Mtb involvement in the liver during the chronic stage of infection.

## PPARγ upregulation in Mtb-infected hepatocytes leads to augmented lipid biogenesis

To understand the molecular mechanism behind the accumulation of lipid droplets in the Mtb-infected liver at 8 weeks post-infection, we investigated the expression patterns of the transcription factors involved in the regulation of the genes of lipid biogenesis in our RNA-seq data. At 48 hr post-infection, the transcript levels of Peroxisome proliferator-activated receptor-gamma (*PPARG*) were upregulated by 5–6-fold. To validate, we performed quantitative real-time PCR analysis of the *PPARG* gene in the infected HepG2 at 48 hr post-infection. With respect to the uninfected control, *PPARG* was upregulated by 3–4-fold. Besides, downstream adipogenic genes that are directly or indirectly controlled by PPARγ protein like *MGAT1, FSP27, FASN, DGAT1, DGAT2, ACAT1, ACAT2, ADIPSIN*, etc were all upregulated by more than twofold in the infected cells. (*Figure 6—figure supplement 1A*). Immunoblot also revealed a greater level of PPARγ protein in primary mouse hepatocytes at 24 hr and 48 hr post-infection in the Mtb-infected cells (*Figure 6C and D*). In the in vivo Mtb aerosol infection model, *Pparg* expression levels showed an intriguing trend, at week 2 post-infection, the expression level was comparable to the uninfected control, while at week 4 post-infection, the expression level spiked to 1.5–2-fold, reaching 3–4 fold at week 8 (*Figure 6B*). The pattern of expression of *Pparγ* in the infected liver correlated with the bacterial load in the liver, where we see the induction at week 4 when Mtb reaches the liver and maximum expression at week 8 when the load of Mtb is considerably high. Besides, PPARγ, some of the critical enzymes involved in fatty acid biosynthesis, TAG biosynthesis, cholesterol esterification like *Fasn, Dgat1, Dgat2, Mgat1, Acat1, Acat2*, etc., showed a temporal increase in expression levels at the later time points post-infection (*Figure 6A*). We then used a specific inhibitor of PPARγ, GW9662 (20 μM), and agonist of PPARγ, rosiglitazone (20 μM) in infected HepG2 and quantified the bacterial load after 48 hr. Importantly, inhibition of PPARγ decreased the bacterial load by almost twofold, while chemically inducing PPARγ increased

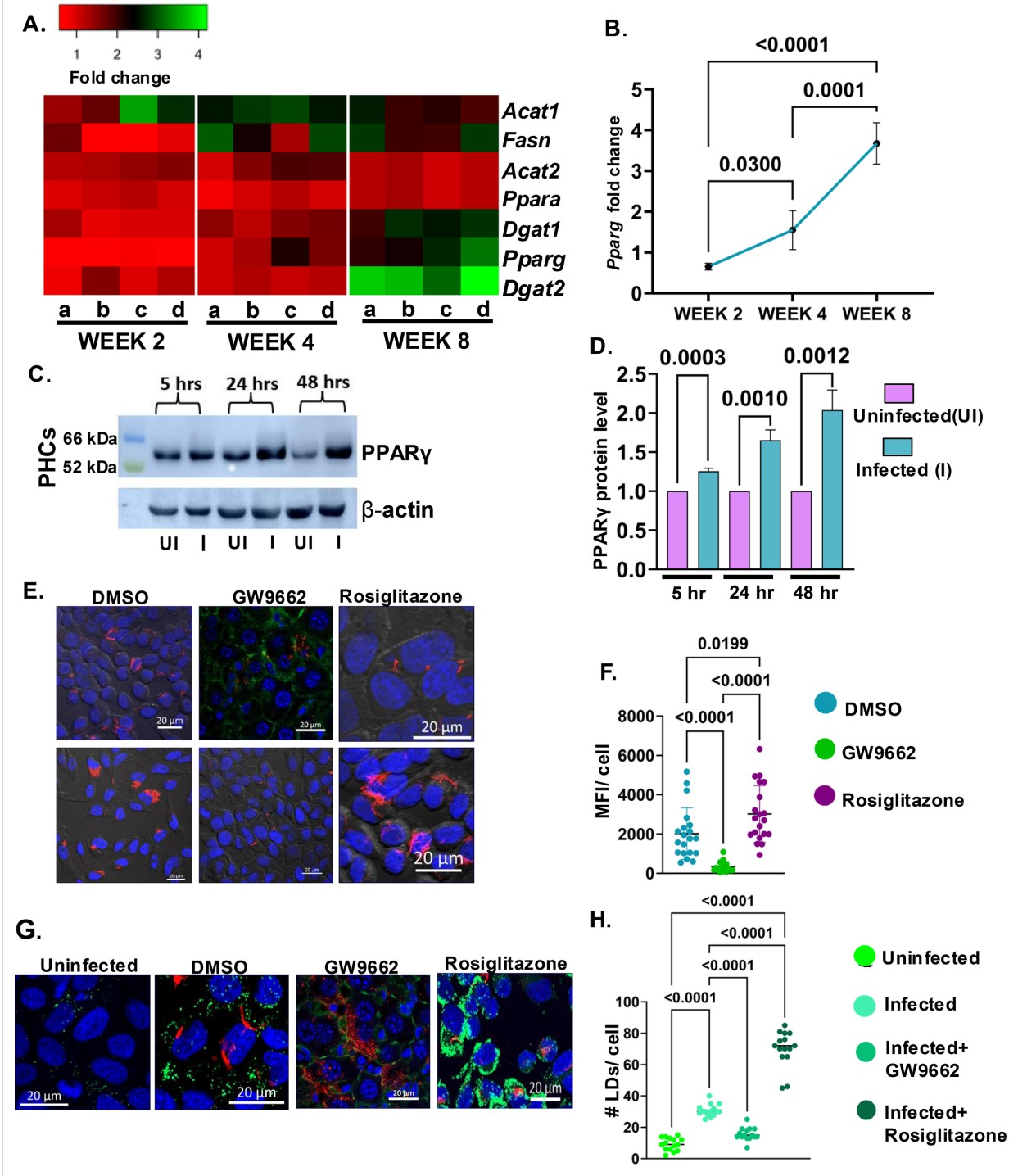

**Figure 6.** Peroxisome proliferator-activated receptor-gamma (PPARγ) driven lipid biogenesis drives *Mycobacterium tuberculosis* (Mtb) growth in hepatocytes. (**A**) Heat map showing the fold change of the genes involved in the lipid biosynthesis and LD biogenesis in liver across weeks 2, 4, and 8, post-infection. (**B**) Kinetic increase in the gene expression of *Pparγ* transcript levels across different weeks post-infection (**C**) Immunoblot showing increased PPARγ protein levels in primary hepatocytes (PHCs) (MOI: 10) at 5-, 24-, and 48 hr post-infection. (**D**) Bar plot showing the increased band intensity of PPARγ in the infected PHCs at the mentioned time points. (**E**) Representative confocal microscopy images showing HepG2 infected with Mtb-mCherry-*H37Rv* and treated with GW9662 and rosiglitazone (**F**) mean fluorescent intensity (MFI)/cell of Mtb-mCherry-*H37Rv* DMSO-treated, GW9662, and rosiglitazone-treated HepG2 cells. (**G**) Representative confocal images of uninfected, infected, GW9662, and rosiglitazone-treated infected HepG2 showing changes in the number of LDs. (**H**) Plot depicting the quantification of LDs/cell in the mentioned conditions. Representative

*Figure 6 continued on next page*

*Figure 6 continued*

data from n=4 biological replicates. Data were analyzed by using the two-tailed unpaired Student's t-test in D and by one-way ANOVA in B, F, and H. *$p<0.05$, **$p<0.005$, ***$p<0.0005$, ****$p<0.0001$, ns = non-significant.

The online version of this article includes the following source data and figure supplement(s) for figure 6:

**Source data 1.** Data used for generating the heatmap in *Figure 6A*.

**Source data 2.** Data used for generating the graph in *Figure 6B*.

**Source data 3.** Labeled and unedited blots shown in *Figure 6C*.

**Source data 4.** Data used for generating the graph in *Figure 6D*.

**Source data 5.** Data used for generating the plot in *Figure 6F*.

**Source data 6.** Data used for generating the plot in *Figure 6H*.

**Figure supplement 1.** *Mycobacterium tuberculosis* (Mtb) induced peroxisome proliferator-activated receptor-gamma (PPARγ) expression in infected hepatocytes.

**Figure supplement 1—source data 1.** Data used for generating the graph in *Figure 6—figure supplement 1A*.

**Figure supplement 1—source data 2.** Data used for generating the graph in *Figure 6—figure supplement 1B*.

**Figure supplement 1—source data 3.** Data used for generating the graph in *Figure 6—figure supplement 1D*.

**Figure supplement 1—source data 4.** Data used for generating the graph in *Figure 6—figure supplement 1E*.

the bacterial load considerably (*Figure 6E and F*, and *Figure 6—figure supplement 1B*). Moreover, LD numbers in cells also directly correlated with the levels of PPARγ (*Figure 6G and H*). To investigate whether *Pparg* expression was also induced in the liver of infected mice, we examined *Pparg* in the liver post-infection. Interestingly, we found enhanced expression PPARγ in the liver of the mice at 8 weeks post-infection (*Figure 6—figure supplement 1C, D*). Moreover, PPARγ protein intensity in hepatocytes was also high in the infected liver (*Figure 6—figure supplement 1E*). Thus, PPARγ activation resulting in lipid droplets formation by Mtb might be a mechanism of prolonging survival within hepatocytes.

## Hepatocyte resident Mtb displays a drug-tolerant phenotype

The success of Mtb as a formidable pathogen depends on its ability to tolerate various anti-TB drugs (*Day et al., 2024*). Drug tolerance is a phenomenon of Mtb surviving drug treatment of longer durations in the absence of any resistance mechanisms. It is a property exhibited by the bacteria but is influenced by both the host and the bacterial factors (*Datta et al., 2024*). Interestingly, hepatocytes are a unique cell type that contains phase I and phase II drug metabolising enzymes (*Almazroo et al., 2017*; *Ahmed et al., 2016*). The pharmacological potency of lipophilic drugs is determined by the rate at which these drugs are metabolized to inactive products.

Cytochrome P450 monooxygenases (cyp450s) system is the key phase I DMEs and known to interact with rifampicin (*Fisher et al., 2009*). We, therefore, analysed whether Mtb infection influences the cyp genes in hepatocytes. *CYP3A4* and *CYP3A43*, respectively, both of which metabolize anti-TB drug rifampicin were upregulated in Mtb-infected HepG2 by approximately fourfold and twofold, respectively (*Figure 7—figure supplement 1A*). Moreover, *NAT2* gene responsible for N-acetylation of isoniazid was also upregulated in the Mtb-infected HepG2 by almost twofold (*Figure 7—figure supplement 1A*). We, therefore, argued that hepatocyte-resident Mtb may display higher tolerance to rifampicin. Towards this, we treated Mtb-infected HepG2 and PHCs with different concentrations of Rifampicin (0.1, 0.5, 5 μg/ml) for 24 hr and CFU enumerated the bacterial after lysis. RAW 264.7 was kept as macrophage control with the similar experimental setup. The percentage of bacteria which survived the drugs was the drug-tolerant population. Both HepG2 and PHCs resident Mtb were significantly tolerant to (25–30%) to rifampicin, as compared to the macrophages (*Figure 7A and B*). Almost 10 % of the bacterial population in HepG2 display a tolerogenic phenotype at the highest antibiotic concentration (*Figure 7B*).

We also examined Mtb susceptibility to isoniazid (INH), another first-line anti-TB drug which is predominantly metabolized (50–90%) via N-acetylation of its hydrazine functionality by arylamine N-acetyltransferase 2 (NAT2) (*Sotsuka et al., 2011*). Interestingly, KEGG analysis of transcriptomic data suggested several genes in this pathway to be upregulated in hepatocytes infected with Mtb (*Figure 7—figure supplement 1B*). Experimental studies indeed showed higher tolerance of Mtb to

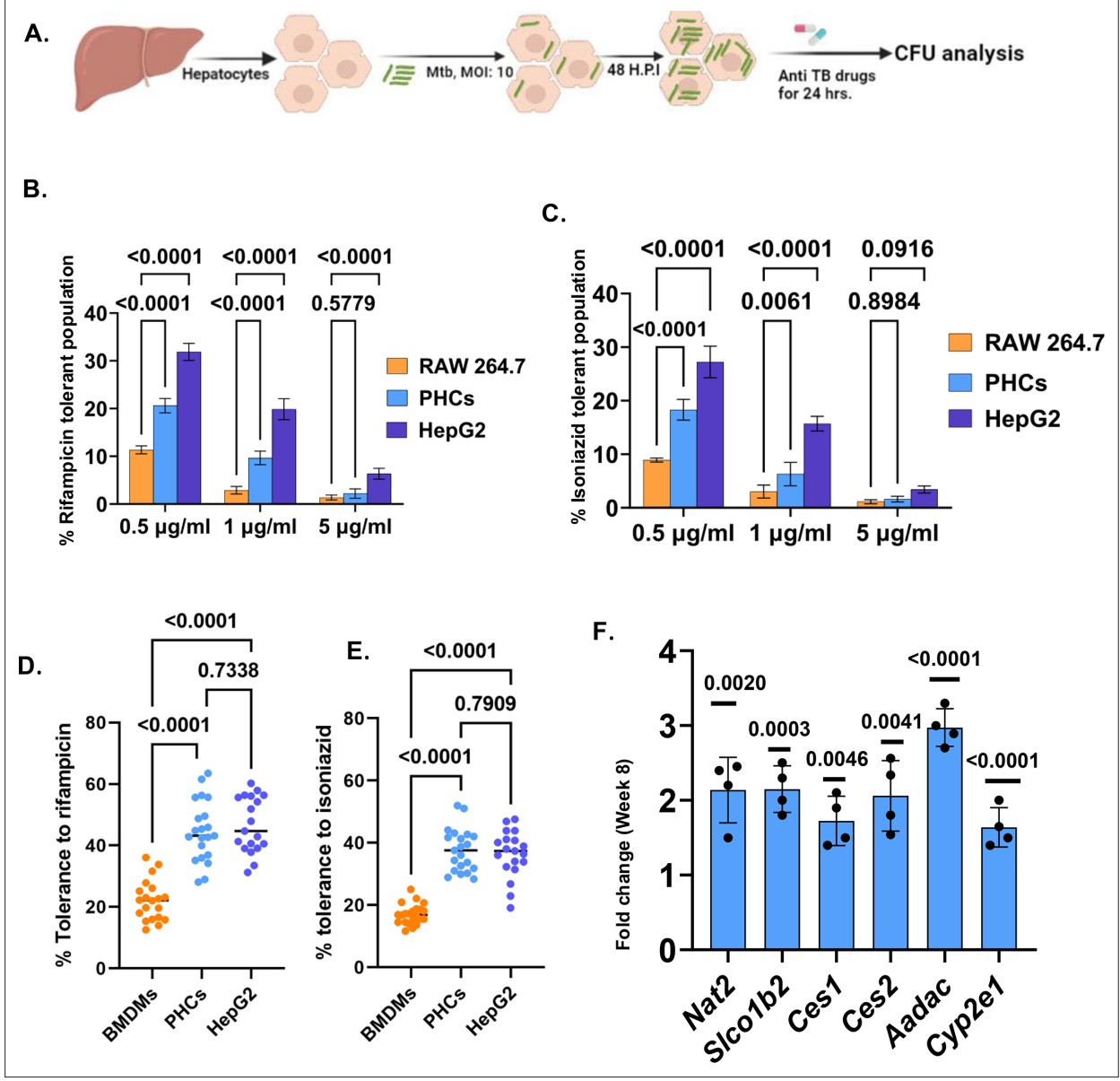

**Figure 7.** Hepatocytes provide a drug-tolerant niche to *Mycobacterium tuberculosis* (Mtb). (**A**) Experimental scheme of deducting the percentage drug tolerance in hepatocytes (generated using Biorender.com) (**B**) Percentage tolerance of Mtb-H37Rv against Rifampicin within RAW 264.7, primary hepatocytes (PHCs), and HepG2 at different time points post-infection. (**C**) Percentage tolerance of Mtb-H37Rv against Isoniazid within RAW 264.7, PHCs, and HepG2 at different time points post-infection. Representative data from three independent experiments. (**D**) Percentage tolerance of Mtb-*H37Rv* against rifampicin within bone marrow-derived macrophages (BMDMs), PHCs, and HepG2 as measured microscopically (**E**) Percentage tolerance of Mtb-H37Rv against isoniazid within BMDMs, PHCs, and HepG2 measured microscopically. Representative data from three independent experiments. Each dot represents a single field having more than four infected cells. 20 such fields were analyzed (**F**) Transcript levels of the various drug modifying enzymes (DMEs) involved in Rifampicin and Isoniazid metabolism in mice liver, 8 weeks post-infection (n=4 mice), the fold change has been calculated by considering the expression in the uninfected mice to be 1. Data were analyzed by using two-way ANOVA in (**B**) and (**C**), one-way ANOVA in (**D**) and (**E**) and the two-tailed unpaired Student's t-test in (**F**) and Representative data from n=4 biologically independent replicates *$p<0.05$, **$p<0.005$, ***$p<0.0005$, ****$p<0.0001$. ns=non-significant.

The online version of this article includes the following source data and figure supplement(s) for figure 7:

**Source data 1.** Data used for generating the graph in *Figure 7B*.

**Source data 2.** Data used for generating the graph in *Figure 7C*.

**Source data 3.** Data used for generating the plot in *Figure 7D*.

**Source data 4.** Data used for generating the plot in *Figure 7E*.

*Figure 7 continued on next page*

*Figure 7 continued*

**Source data 5.** Data used for generating the graph in *Figure 7F*.

**Figure supplement 1.** Gene expression analysis of key drug modifying enzymes (DMEs) related to rifampicin and isoniazid metabolism.

**Figure supplement 1—source data 1.** Data used for generating the graph in *Figure 7—figure supplement 1A*.

**Figure supplement 2.** Microscopic images of *Mycobacterium tuberculosis* (Mtb)-infected bone marrow-derived macrophages (BMDMs), primary hepatocytes (PHCs), and HepG2 treated with rifampicin and isoniazid.

**Figure supplement 3.** *Mycobacterium tuberculosis* (Mtb) derived from hepatocytes show no changes in drug efflux pumps and antibiotic sensitivity.

**Figure supplement 3—source data 1.** Data used for generating the graph in *Figure 7—figure supplement 3C*.

**Figure supplement 3—source data 2.** Data used for generating the graph in *Figure 7—figure supplement 3D*.

INH in both primary hepatocytes and HepG2 at different concentrations (0.1, 0.5, 5 µg/ml) (*Figure 7A and C*). To corroborate the CFU data, we repeated the same experiment (*Figure 7A*) and looked at the bacterial load within each of murine PHCs and BMDMs using high-resolution confocal microscopy. Mouse BMDMs were used as a macrophage control. Here also we have calculated the percentage tolerance as a ratio by measuring the mean fluorescent intensity of GFP-Mtb per hepatocyte treated with drug to MFI of GFP-Mtb per hepatocyte treated with DMSO (control). Extensive analysis of more than 200 cells distinctly showed the presence of more Mtb within PHCs treated with both rifampicin and isoniazid than in BMDMs treated with the same concentration of the drugs (*Figure 7D and E* and *Figure 7—figure supplement 2A*).

To check whether some of these key DMEs are also altered in the Mtb-infected mice liver, we did qRT-PCR of some key DMEs related to both rifampicin and isoniazid metabolism. Transcript levels of some of the key DMEs were upregulated in the liver of the infected mice, 8 weeks post-infection like *Nat2, Cyp2e1, Slco1b2, Ces1, Ces2,* and *Aadac* (upregulated by almost 2–3-fold) (*Figure 7F*).

Drug tolerance is a bacterial property influenced by both bacterial and host-induced factors. To check whether Mtb derived from hepatocytes, post-drug exposure has altered drug sensitivity or not, we conducted a resazurin microtitre assay with different dilutions of both rifampicin and isoniazid with Mtb derived from PHCs treated with DMSO, Mtb derived from PHCs post-exposure to rifampicin and Mtb derived from PHCs post-exposure to isoniazid. Mtb from all three different setups did not show any alterations in their MIC values, displaying no change in drug sensitivity (*Figure 7—figure supplement 3A*). The increased tolerance of Mtb within the hepatocytes can also be attributed to the increased activity of the drug efflux pumps. Drug efflux pumps like Rv1258c, Rv1410, and Rv1819c are known to get upregulated under drug stress conditions, preventing the accumulation of the drugs and thereby decreasing drug sensitivity. To this end, we isolated intracellular Mtb RNA from HepG2 treated with DMSO, 0.5 µg/ml rifampicin, and 0.5 µg/ml isoniazid and checked for the expression of key drug efflux pumps like *Rv 1258 c. Rv1410, Rv1819* etc in these three Mtb populations (*Figure 7—figure supplement 3B, C and D*). Interestingly, quantitative RT-PCR analyses of these Mtb genes did not show any upregulation of these transporters in Mtb derived from HepG2 exposed to rifampicin and isoniazid, indicating that drug tolerance phenotype can be better attributed to host intrinsic factors rather than Mtb efflux pumps. Our data show that Mtb-infection of hepatocytes induces DMEs, and this host-specific extrinsic activation may result in decreased bioavailability or increased inactivation of anti-TB drugs.

## Discussion

Over the years, various anatomical and cellular environments conducive to Mtb infection have been identified, particularly concerning its latency (*Cosma et al., 2003*; *Bussi and Gutierrez, 2019*). Pulmonary TB typically presents with classical symptoms such as persistent coughing and mucus production due to lung involvement (*Luies and du Preez, 2020*). While systemic manifestations like weight loss, fatigue, and loss of appetite are commonly linked to TB progression, the underlying factors driving these metabolic derangements remain unexplored. Our study sheds light on hepatocytes as a previously unrecognized site for Mtb survival and replication, thereby extrapolating the current understanding of TB beyond its pulmonary focus. By combining in vitro, ex vivo, in vivo, and clinical data, we demonstrate that Mtb not only persists within hepatocytes but also induces important metabolic

reprogramming, with significant implications for disease progression, symptomatic manifestations, and drug tolerance.

A central finding of our work is the Mtb-mediated activation of peroxisome proliferator-activated receptor gamma (PPARγ) in hepatocytes. PPARγ activation leads to the accumulation of cholesterol esters (CE 16:0, 18:0, 18:1), diacylglycerols (DAGs 36:1, 36:2, 34:1), and triacylglycerols (TAGs 18:1/36:2, 18:0/36:2, 18:0/36:1). These lipid pools contribute to the formation of lipid droplets, which colocalize with intracellular Mtb, serving as a nutrient reservoir that facilitates bacterial persistence and proliferation. The upregulation of key TAG biosynthesis genes such as *Tgs1*, *Tgs4*, and *Rv1760* in Mtb residing within hepatocytes underscores a transcriptional rewiring within the pathogen to assimilate lipids as a source of their nutrients. Our pharmacological studies using PPARγ agonist and antagonist demonstrated that modulation of PPARγ directly impacts Mtb burden in hepatocytes, confirming the pivotal role of PPARγ in Mtb survival in hepatocytes. The lipid remodeling induced by Mtb infection in hepatocytes is recapitulated in the murine aerosol model, where an increased number of lipid droplets was observed at week 8, accompanied by localized accumulation of immune cells and granuloma-like structures. These findings are consistent with previous reports in macrophages where PPARγ activation enhances lipid biosynthesis, regulates immune responses, and impedes host cell apoptosis (*Arnett et al., 2018*; *Rajaram et al., 2010*). Recent findings have further underscored heightened levels of PPARγ in peripheral blood mononuclear cells (PBMCs) from TB patients with elevated cortisol levels and increased disease severity (*Díaz et al., 2023*). However, hepatocytes are uniquely positioned in systemic lipid regulation, engaging in de novo fatty acid synthesis, TAG synthesis, β-oxidation, and lipoprotein metabolism (*Zhang et al., 2022*; *Bechmann et al., 2012*). Mtb-mediated perturbation of these vital processes can lead to a sequela of metabolic disorders such as non-alcoholic fatty liver disease (NAFLD), dyslipidemia, and insulin resistance. In macrophages, heightened intracellular lipid levels have been observed to impede autophagy and the acidification of phagolysosomes, both crucial for bacterial eradication (*Lovewell et al., 2016*). Despite these parallels, comparative transcriptomic analyses of infected HepG2 cells with THP1 cells highlight differences in pathways such as vacuolar and vesicular transport, xenobiotic metabolism, macroautophagy, and cellular respiration. Furthermore, hepatocytes secrete hepatokines, a class of proteins serving as signaling molecules, with diverse roles in metabolism, inflammation, and energy homeostasis, thereby influencing the host-pathogen interplay (*Jensen-Cody and Potthoff, 2021*). Intriguingly, a recent investigation demonstrated that hepatic PPARγ activation induces the expression of growth differentiation factor 15 (GDF-15), a crucial regulator of weight loss observed in ketogenic diets (*Lu et al., 2024*).

Additionally, we suggest that Mtb infection of hepatocytes creates a drug-tolerant environment in the liver due to activation of DMEs, many of which are earlier shown to metabolize the two frontline drugs, isoniazid and rifampicin. Moreover, lipid accumulation in the liver might also indirectly alter the levels of drug metabolizing enzymes as reported in previous studies (*Fisher et al., 2009*; *Rey-Bedon et al., 2022*). Upregulation of transcripts is not only observed in infected cells in vitro, but also in the livers of mice eight weeks after Mtb infection. Particularly noteworthy is the upregulation of *NAT-2*, which controls the rate-limiting step of acetylating isoniazid into acetylisoniazid. This metabolite is further processed into acetylhydrazine and isonicotinic acid, ultimately reducing the drug's effectiveness against Mtb. Similarly, key rifampicin-metabolizing esterase genes such as *Ces1*, *Ces2*, *Aadac*, and transporter *Slco1b2* exhibit upregulation, potentially influencing the drug's distribution and metabolism in the body. The activation of DMEs can significantly modify the pharmacokinetics and pharmacodynamics of anti-TB medications, resulting in suboptimal drug concentrations and the emergence of drug-resistant strains (*Dookie et al., 2018*).

Pulmonary TB can lead to miliary through hematogenous dissemination, where Mtb spreads from the infected lungs into blood vessels either from a primary lung focus, reactivated TB, or caseous necrosis. Once in blood vessels, the bacteria seed multiple organs, forming tiny granulomas, characteristic of miliary TB. The liver becomes involved either through direct hematogenous spread or extension from nearby infected lymph nodes, leading to hepatic TB, which presents with granulomas and liver dysfunction. This systemic spread underscore the severity of untreated pulmonary TB and the need for early intervention (*Shiloh, 2016*; *Coleman et al., 2022*; *Wei et al., 2024*; *McMullan and Lewis, 2017*). Our murine and guinea pig aerosol infection models demonstrated progressive liver involvement starting at week 4, with the presence of lipid-laden hepatocytes, localized immune infiltrates, and granuloma-like structures. Importantly, liver biopsy samples from TB patients revealed

ectopic granulomas and Mtb antigen (Ag85B) localization within hepatocytes, highlighting the clinical relevance of our findings. While previous research has explored liver infection by Bacillus Calmette-Guérin (BCG) using an intravenous model in mice, the bacilli were predominantly found within tissue-resident macrophages or Kupffer cells at 15 min and 2 days post-infection (*Seiler et al., 2001*). However, our study consistently identified the presence of Mtb within hepatocytes in a murine aerosol infection model after the fourth week. Similarly, in a guinea pig aerosol infection model, liver infection was evident, marked by distinct granulomas by week 4. Furthermore, analysis of human biopsy liver samples from pulmonary TB patients revealed ectopic granuloma-like structures within the liver hepatocytes, with the presence of Mtb-specific Ag85B signals within hepatocytes. Together, these findings underscore hepatocytes as a novel niche for Mtb persistence, shedding new light on the pathogenesis of TB. Interestingly, in infected armadillos, *M. leprae* is shown to infect hepatocytes and hijack liver homeostatic, regeneration pathways to promote de novo organogenesis (*Hess et al., 2022*).

Although animal models offer valuable insights into Mtb infection and TB pathology, they fall short of fully replicating the complexities of human TB. Several recent studies in COVID-19 patients have shown that hepatocytes get infected with SARs-CoV-2, thereby increasing gluconeogenesis and hyperglycemia (*Barreto et al., 2023*; *Mercado-Gómez et al., 2022*). With compelling reports associating hepatic steatosis with the onset of type 2 diabetes mellitus (T2DM), we speculate that TB-induced perturbations in lipid metabolism might predispose chronic TB patients to T2DM and vice versa (*Hazlehurst et al., 2016*; *Dharmalingam and Yamasandhi, 2018*). Through our investigation, we propose that future studies in human TB patients might scrutinize this metabolically rich hepatocyte niche to understand multiple organ-wide derangements in TB pathogenesis.

## Materials and methods

### Confocal microscopy and Immunofluorescence measurements

Lipid droplets were stained with BODIPY 493/503 dye. Primary mouse hepatocytes and HepG2 cells were grown on 12 mm coverslips at respective densities as per the experimental requirement. Cells were washed with 1X PBS (HiMedia M1452-500G), fixed with 4% paraformaldehyde, and incubated at room temperature for 15 min. After incubation, the cells were washed thrice with 1X PBS followed by staining with 10 μM BODIPY dye in 1X PBS for 45 min at room temperature. After the staining, the excess dye was removed by washing thrice with 1X PBS. To check for the acidified compartments within the cells, LysoTracker Red DND-99 (L7528) was added to the cells at a concentration of 500 nM for 30 min at 37°C, followed by washing thrice with 1X PBS to remove the residual dye. The cells were fixed with 4% paraformaldehyde as previously mentioned. For staining with various antibodies, the cells were permeabilized with 0.2% Triton-x 100 (X-100-1L) for 15 min, followed by proper washing with 1X PBS. The cells were then blocked by 2% BSA in 1X PBST for 1 hr at room temperature. Post blocking, the cells were treated with primary antibody overnight at 4°Celsius, followed by proper washing with 1X PBS. Secondary antibody (1:500 dilution) was added to the cells for 1 hr at room temperature. Three washes with 1X PBS were given. The nucleus was stained with DAPI (Sigma D9542-5MG) at 1ug/ml concentration for 20 min at room temperature. The excess DAPI was washed with 1X PBS followed by mounting with ProLong Gold Antifade mountant (P36930). Images were acquired by Zeiss LSM 980 Laser scanning confocal microscope.

### Image analysis

Analysis was done using Image J (RRID:SCR_003070) and Zeiss Zen Microscopy software (RRID:SCR_013672). Mean fluorescent intensity/ cell was calculated by corrected total cell fluorescence (CTCF) = Integrated Density – (Area of Selected Cell x Mean Fluorescence of Background readings). Signal intensity in tissue sections were normalized to respective areas. For percentage tolerance of Mtb within hepatocytes, we have calculated the ratio by measuring the mean fluorescent intensity of GFP-Mtb per hepatocyte treated with drug to MFI of GFP-Mtb per hepatocyte treated with DMSO (control). BMDMs were used as the macrophage control. More than 20 fields, each consisting of more than four infected cells have been used for analysis. MFI/ cell was calculated using the Zeiss ZEN blue image analysis software.

### Tissue immunofluorescence staining

#### Paraffin sections

Five-micron thick sections of paraffin-embedded tissue sections were taken in poly-L-lysine-coated slides (P0425-72EA). Deparaffinization was performed by heating the slides at 50°C for 20 s (three times) till the wax melts, followed by the subsequent steps, 100% xylene (Merck, CAS-1330-20-7) for 10 min (three times), xylene and absolute ethanol (Merck, CAS- 64-17-5) for 10 min, 100% ethanol for 10 min, 70% ethanol for 5 min (two times), 50% ethanol, distilled water for 5 min (2 min) and a final wash in 1x PBS for 5 min (two times). Antigen retrieval was performed in an antigen retrieval buffer (10mM Sodium Citrate, 0.05% Tween-20, pH: 6) by heating the slides at 60°C for 15 min. After antigen retrieval, permeabilization was performed with 0.4% Triton-X 100 in 1X PBS for 20 min followed by proper washing with 1x PBS. Blocking was done with 5% BSA for 1 hr. Sequential addition of primary antibody was performed at 1:100 dilution at 4°C overnight. Primary antibody was washed with 1X PBS followed by counterstaining with DAPI nuclear stain at 1 µg/ml concentration. Mounting was done with a drop of Vectashield (Sigma-Aldrich, F6182-20ml). The slides were visualised in Zeis LSM 980 confocal microscopy at 40X (oil) magnification.

#### Cryosections

Seven-micron thick cryosections were taken in poly-L-Lysine-coated slides (P0425-72EA). The sections were washed with 1X PBS, three times for 5 min each. Permeabilisation was done with 0.25% Triton-X 100 in 1X PBS for 15 min followed by washing with 1X PBS. Blocking was done with 5% BSA for 1 hr followed by incubation with primary antibody overnight at 4°C. After that, the slides were washed with 1X PBS 2 times for 5 min each followed by incubation with fluorophore-conjugated secondary antibody for 45 min. The slides were washed with 1X PBS followed by counterstain with DAPI at 1 µg/ml concentration. Mounting was done with a drop of Vectashield (Sigma-Aldrich, F6182-20 ml). The dilution of the dyes and antibodies used in the staining are mentioned in the table.

For staining with BODIPY, the sections were incubated with 15 µM of BODIPY for 40 min at room temperature. The slides were visualized in Zeiss LSM 980 confocal microscopy at 40X (oil) magnification.

### Fluorescence in situ hybridization

FISH was used to detect Mtb in infected human liver following published protocols (*World Health Organisation, 2023*; *Seung et al., 2015*; *Husain et al., 2016*). Briefly, the paraffinized human liver tissue sections were initially deparaffinized using a serial washing step with xylene and ethanol, following which the sections were treated with 1 mg/ml Proteinase K and 10 mg/ml Lysozyme in 10 mM Tris (pH 7.5) at 37°C for 30 min. Next, the samples were incubated in the prehybridization buffer at 37°C for 1 hr. Prehybridization buffer is composed of 20% 2X Saline sodium citrate (SSC), 20% Dextran sulfate, 30% Formamide, 1% 50X Denhardt's reagent, 2.5% of 10 mg/ml PolyA, 2.5% of 10 mg/ml salmon sperm DNA, 2.5% of 10 mg/ml tRNA. The slides were thoroughly washed with a 2X SSC buffer. The sections were then incubated in hybridization buffer at 95°C for 10 min and then chilled on ice for 10 min. Further hybridization was allowed at 37°C overnight. Hybridization buffer is composed of prehybridization buffer plus 16S Mtb-$H_{37}$ Rv probe (5' FITC – CCACACCGCTAAAG – 3'), which is specific for the 16S rRNA of Mtb at a final concentration of 1 ng/µl. The liver tissue sections were next subjected to a series of washing steps with 1 X SSC at room temperature for 1 min, 1 X SSC at 55 °C for 15 min, 1 X SSC at 55 °C for 15 min, 0.5 X SSC at 55 °C for 15 min, 0.5 X SSC at 55 °C for 15 min, 0.5 X SSC at room temperature for 10 min. Coverslips were mounted on glass slides and visualized using Nikon A1R confocal microscope with a 488 nm laser.

#### Acid-fast staining and Auramine O and Rhodamine B staining in liver sections

Acid-fast staining was performed using a ZN Acid Fast Stains-Kit (K005L-1KT, HIMEDIA). Prior to staining, the paraffinized samples were deparaffinized using a serial washing step with xylene and ethanol (1) 100% Xylene for 6 min, (2) Xylene: Ethanol 1:1 for 3 min, (3) 100% Ethanol for 3 min, (4) 95% Ethanol for 3 min, (5) 70% Ethanol for 3 min, (6) 50% Ethanol for 3 min, (7) Distilled water. The glass slides were flooded with Carbol Fuchsin stain and heated to steam for 5 min with a low flame. The glass slides were allowed to stand for 5 min without further heating. The glass slides were then

washed in running tap water. The glass slides were decolorized with acid-fast decolorizer for 2 min. (5) Washed with tap water. (6) Counterstain for 30 s with Methylene Blue washed with tap water, dried in air, and examined under 100x objective with oil immersion. The presence or absence of bacteria in infected and uninfected samples was checked through staining with Phenolic Auramine O-Rhodamine B dye (*Eckhardt et al., 2020*) (1/3 dilution of stock solution) (Auramine O-861020-25gm, Sigma) (Rhodamine B-R6626-100gm, Sigma). Coverslips were mounted on glass slides and visualized using CLSM with a 488 nm laser.

## Bacterial cultures and in vitro experiments

Virulent laboratory strains of H37Rv, BCG, and GFP-H37Rv bacterial cultures were cultivated on 7H9 medium (BD Difco) supplemented with 10% Oleic Acid-Albumin-Dextrose-Catalase (OADC, BD, Difco), 0.05% glycerol, and 0.05% Tween 80 under shaking at 37°C conditions for in vitro assays. The cultures were then incubated in an orbital shaker at 100 rpm and 37 °C until the mid-log phase. pMN437-GFPm2 vector (Addgene, 32362) was used to electroporate the virulent H37Rv strain to create GFP-H37Rv, which was then maintained in 50 µg/ml hygromycin 7H9-OADC medium. pMSP12:mCherry plasmid (RRID:Addgene_30169) was electroporated in H37Rv to generate Mtb-H37Rv-mCherry. To prepare the single-cell suspension needed for infection tests, bacterial cultures were passed through a sequence of different gauge needles five times through 23-gauge, 26-gauge, and three times through 30-gauge.

Human monocytic cell line THP-1 were obtained from American Type Culture Collection (ATCC) and cultured in RPMI-1640 medium with 10% FBS at 37°C and 5% CO2 incubator. THP-1-derived macrophages were obtained by incubating THP-1 cells with 20 ng/ml phorbol 12-myristate 13-acetate (PMA, sigma) for 24 hr followed by washing and maintenance in complete media. The cells were kept in non-PMA-containing complete media for 16-20 hr before infection with Mtb. HepG2, Huh-7, RAW 264.7, and PHCS were grown in DMEM with 10% FBS at 37°C and 5% $CO_2$ incubator.

AML-12 cells were cultured in DMEM media containing 1X insulin-transferrin-selenium supplement (ITS-G), under the above-mentioned conditions.

In vitro and ex vivo infection experiments in primary cells and different cell lines were performed at a multiplicity of infection of 10 (MOI: 10) for both CFU enumeration and confocal microscopy. The macrophage experiments involving THP-1 and RAW 264.7 involved incubating the cells with Mtb-H37Rv for 5-6 hr followed by washing with 1X PBS and amikacin treatment (200 µg/ml for 2 hr) to remove the extracellular bacteria. The cells were kept for the designated time points for 24 hr, 48 hr, and so on and then lysed with lysis buffer (0.05 % SDS in 1X PBS) followed by plating in 7H11 plates.

For primary mouse hepatocytes and AML-12 cells, the cells were infected with the Mtb at a multiplicity of 10 for 8 hr followed by amikacin treatment (200 µg/ml for 2 r). For HepG2 and Huh-7, the time of incubation was 5- 6 hr followed by washing with 1X PBS and amikacin treatment as previously mentioned to remove the extracellular bacteria.

HepG2 (RRID:CVCL_0027), THP-1 (RRID:CVCL_0006), AML-12 (RRID:CVCL_0140), and RAW 264.7 (CVCL_0493) were obtained from the ATCC (American Type Culture Collection).

All the cell lines were authenticated by STR (short tandem repeats profiling), and the different cells were periodically checked for mycoplasma contamination with PCR-based methods and DAPI staining. For HepG2, further cell type confirmation was carried out by looking at the expression of the albumin transcript by quantitative real-time PCR and albumin protein expression by confocal microscopy, both of which showed robust levels.

The percentage drug-tolerant population was calculated using the following formula (A/B X 100) where A is the CFU in the drug-treated group, B is the CFU in the untreated group. The cells were morphologically checked for signs of cell death before proceeding with the plating. For the inhibitor and inducer experiments, the cells were treated with the respective drugs for 48 hr post uptake of Mtb. After that, they were lysed and Mtb CFU was enumerated.

## Standardization of MOI for PHCs and HepG2

To standardise the MOI, HepG2 and BMDMs were infected with the following MOI of GFP-Mtb as per the standardized protocol: 1, 2.5, 5, and 10. 6 hr post-infection, the cells were trypsinized and fixed with 2% PFA. The percentage uptake was calculated by flow cytometry. For PHCs, the cells were

infected at the above-mentioned MOI and percentage infection was calculated by microscopy as (# of cells with GFP-Mtb/Total # of cells *100).

## C57BL/6 aerosol challenge

The mice infection experiments were conducted in the Tuberculosis Aerosol Challenge Facility (TACF, ICGEB, New Delhi, India). C57BL/6 mice were placed in individual ventilated cages within the enclosure, maintaining a temperature of 20–25°C, 30–60% humidity, and 12 hr-12 hr of light-dark cycle. Following the standardized protocol, mice were infected with 200 CFUs of H37Rv in a Wisconsin-Madison chamber. To ensure proper establishment of infection, two animals were euthanized 24 hr post-aerosol challenge. The lungs were harvested and homogenised in 1X PBS and plated in Middlebrook 7H11 agar plates (Difco) supplemented with 10% OADC and 0.5% glycerol. CFU enumeration was done three weeks post-plating.

## C57BL/6 peritoneal infection

The mice were injected with $10^6$ CFUs of H37Rv in 0.2 ml of 1X PBS. Following infection, on different days post-infection, the lung, spleen, and the liver was harvested and homogenized and plated in Middlebrook 7H11 agar plates (Difco) supplemented with 10% OADC and 0.5% glycerol. CFU enumeration was done three weeks post-plating.

## List of antibodies and dyes used in the study

| Serial number | Antibodies and dyes | Catalogue number and RRID | Company | Application and dilution |
|---|---|---|---|---|
| 1. | Rab5 | Cat#:2143T, RRID:AB_823625 | CST | IF/ICC (1:200) |
| 2. | Rab7 | Cat#: 9367T, RRID:AB1904103 | CST | IF/ICC (1:200) |
| 3. | Cathepsin D | Cat#: Ab75852, RRID:AB_152367 | Abcam | IF/ICC (1:150) |
| 4. | LAMP1 | Cat#: 9091T, RRID:AB_2687579 | CST | IF/ICC (1:200) |
| 5. | EEA1 | Cat#: 3288T, RRID:AB_2096811 | CST | IF/ICC (1:200) |
| 6. | ASPGR1 | Cat#:PA5-32030, RRID:AB_2549503 | Invitrogen | IF/ICC (1:500) |
| 7. | PPAR γ | Cat#:PA3-821A, RRID:AB_2166056 | Invitrogen | IF/ICC (1:150) WB (1:1000) |
| 5. | Ag85B | Cat#:Ab43019, RRID:AB_776575 | Abcam | IF/ICC (1:800) |
| 6 | CD45-APC (clone 104) | Cat#:558702, RRID:AB_1645215 | BD biosciences | 1:300 |
| 7. | β-actin rabbit polyclonal Antibody. | Cat#:4967 S RRID:AB_330288 | CST | IF/ICC (1:100) WB (1:2000) |
| 8. | Goat anti-rabbit IgG (H+L) secondary Ab Alexa Fluor 555 Plus | Cat#:A32732, RRID:2633281 | Thermo Fisher Scientific | IF/ICC (1:400) |
| 9. | Goat anti-rabbit IgG (H+L) Secondary Antibody, Alexa Fluor 488 | Cat#:A11008, RRID:AB_143165 | Thermo Fisher Scientific | IF/ICC (1:400) |
| 10. | Goat anti-rabbit IgG (H+L) Secondary Antibody, Alexa Fluor 405 | Cat#:A-31556, RRID:AB_221605 | Thermo Fisher Scientific | IF/ICC (1:400) |
| 11 | Anti-Albumin antibody | Cat#:PA5-143811 RRID:AB_3075025 | Thermo Fisher Scientific | IF/ICC (1:250) |
| 12 | Chicken anti-rabbit IgG (H+L) Secondary antibody Alexa Fluor 647 | Cat#:A-21443 RRID:AB_2565861 | | IF/ICC (1:1000) |

*Continued on next page*

*Continued*

| Serial number | Antibodies and dyes | Catalogue number and RRID | Company | Application and dilution |
|---|---|---|---|---|
| 13. | Lysotracker Red | L7528 | Invitrogen | 500 nM |
| 14. | BODIPY-493/503 | D3922 | Invitrogen | Staining (10–15 µM) |
| 15. | BODIPY 558/568 C12 | D3835 | Invitrogen | Metabolic Labeling (7.5 µg/ml) |
| 16. | Phalloidin-Alexa Fluor 488 | A12379 | Invitrogen | IF/ICC 1 unit/slide |
| 17. | Phalloidin-Alexa Fluor 555 plus | A30106 | Invitrogen | IF/ICC 1 unit/slide |
| 18. | HCS LipidTOX Red Neutral Lipid Stain | H34476 | Invitrogen | IF/ICC (1:400) |
| 17. | DAPI | D9542 | Sigma-Aldrich | 1 ug/ml |
| 18. | ZN Acid Fast Stains | K005L-1KT | Himedia | |
| 19. | Auramine O | 861020–25 gm | Sigma-Aldrich | |
| 19 | Rhodamine B | R6626-100gm | Sigma-Aldrich | |

## List of reagents in the study

| Serial no | Reagents | Catalogue number |
|---|---|---|
| 1. | C75 | C5490-5MG, Sigma-Aldrich |
| 2. | OA-BSA | O3008-5ML, Sigma-Aldrich |
| 5. | GW9662 | M6191-25MG, Sigma-Aldrich |
| 6. | Rosiglitazone | R2408-10MG, Sigma-Aldrich |
| 7. | T863 | SML0539-5MG, Sigma-Aldrich |
| 8. | Rifampicin | 557303–1 G, Merck |
| 9. | Isoniazid | I3377-5G, Merck |
| 10. | PMA (Phorbol 12-myristate 13-acetate) | P8139-1MG, Merck |
| 11. | Mouse MCSF (macrophage colony-stimulating factor) | 130-101-706 (25 µg), Miltenyi Biotec |
| 12. | Insulin-Transferrin-Selenium (ITS -G) (100 X) | 41400045, Gibco |

## Sorting of labeled H37Rv cells

The infected hepatocytes were trypsinized and washed with 1X PBS followed by passing through 40 µm cell strainer to make single cell suspension. After that, the cells positive for mCherry signals were sorted using the BD FACS Aria at TACF, ICGEB. Approximately 1 million cells were used for RNA isolation per sample.

## Primary hepatocyte isolation

After euthanizing the mice, the peritoneal cavity was opened, and the liver was perfused with 1X HBSS solution till exsanguination was complete. After that collagenase solution (17 mg in 30 ml of 1X HBSS) was passed for digestion. The liver was cut into small pieces and kept in 25 ml of DMEM media followed by lysing the tissues with a glass pestle and sieve. The cells were then passed through a 70 µM cell strainer. 25 ml of Percoll was added to the cells with proper mixing. It was centrifuged at 1000 rpm for 5 min. The floating cells were removed, and a brownish layer of pure hepatocytes was pelleted at the bottom. The cells were counted and plated according to the experimental need on a collagen-coated plate.

## Isolation of BMDMs from mouse

BMDMs used for Mtb infection studies were isolated following the published protocol with minor modifications (*Toda, 2021*). Briefly, the epiphysis of the femurs and tibia from C57BL/6 mice were cut

and the bone marrow were gently flushed into BMDM supplemented with 10% FBS and 1% penicillin streptomycin. The bone marrow cells were centrifuged at 200 X g for 5 min at 4°C. The cell pellet was aspirated with 1X PBS and treated with RBC Lysis buffer for 3 min on ice, followed by the addition of complete media. The cells were resuspended in 5 ml of the media and passed through a 70 µm cell strainer. The cells were seeded in complete DMEM supplemented with 10 ng/ml of MCSF. On the third day, half of the media volume was replaced by fresh DMEM supplemented with 10ng/ml of MCSF. On the seventh day, the cells were trypsinized, counted, and seeded according to experimental requirements.

## Lipid extraction protocol and mass spectrometry

Five million HepG2 cells were infected with Mtb-H37Rv at MOI 10 and kept for 24 hr and 48 hr post-infection. An equal number of uninfected cells were taken. The cells were scrapped, and the procedure of Bligh and Dyer was followed *Bligh and Dyer, 1959*. In brief, the cells were lysed in 1% Triton X-100 after being rinsed twice with 1X PBS. Following lysis, the lysate was vortexed and four volumes of methanol-chloroform (2:1) were added. After that, one volume each of water, chloroform, and 50 mM citric acid was added and vortexed. Following a 10-min centrifugation at 10,000 rpm at 4°C, the lower organic phase was collected and dried using liquid nitrogen. All semi-quantitative lipid measurements were done using previously reported high-resolution MS/MS methods and chromatographic techniques on an Agilent 6545 QTOF instrument. All sterols were resolved using a Gemini 5U C-18 column (Phenomenex) while DAGs/TAGs were resolved using a Luna 5U C-5 column (Phenomenex) using established solvent systems.

## Cell lysate preparation and Western blotting

The cells were washed twice with 1X PBS followed by lysis with SDS-RIPA buffer (50 mM Tris-HCl pH 7.5, 150 mM NaCl, 1 mM EDTA, 0.1% SDS, 1% Triton-X 100, 1 mM DTT, 1X Proteinase inhibitor). The cells were incubated with the buffer for 30 min in ice followed by vortexing for 5 min. The supernatant fraction was collected by centrifuging at 10,000 rpm for 20 min at 4°C. The protein concentration was determined by bicinchoninic acid (BCA) protein estimation kit (Thermo Fisher Scientific, Waltham, Massachusetts, USA, 23227) following the manufacturer's protocol. 60-80 ug of protein was resolved in SDS-PAGE followed by transferring onto a PVDF membrane. Blocking was done in 5% skimmed milk in 1X TBST followed by incubation with primary antibodies overnight at 4°C. The membranes were washed three times with 1x TBST for 10 min each followed by incubation with the HRP-conjugated secondary antibody for 1 hr. Immobilon HRP substrate was used to develop the blots and ImageQuant Chemiluminescent imaging system (LAS 500) (RRID:SCR_002798) was used to acquire the images. Band intensities were measured by using ImageJ (RRID:SCR_003070).

## MIC determination of Mtb isolated from primary hepatocytes and primary hepatocytes treated with rifampicin and isoniazid

Mtb was isolated from primary hepatocytes, primary hepatocytes treated with 0.5 µg/ml of rifampicin and primary hepatocytes treated with 0.5 µg/ml of isoniazid. These bacteria were grown to a logarithmic phase of $OD_{600} = 0.6$ in Middlebrook 7H9 broth supplemented with 10% OADC, 0.05% glycerol, and 0.05% Tween 80 under shaking conditions. Single-cell suspension of the Mtb was made as previously described. Twofold serial dilutions of both rifampicin and isoniazid was prepared in 0.1 ml 7H9-OADC (without Tween 80) (The concentration and the dilution of the drugs have been mentioned in the figure) in 96-well flat bottom microplates. Approximately $5×10^4$ Mtb cells were added to each well in a volume of 0.1 ml. Control wells containing (Mtb only, medium + inhibitor, and only medium) were included in the plate setup. The plate was incubated at 37°C 5 days, followed by the addition of 20 µl of 0.02% resazurin for 24 hr. Visually, minimum inhibitory concentration (MIC) was noted as the lowest drug concentration that prevented the change of color from blue to pink.

## qRT-PCR of intracellular Mtb isolated from hepatocytes

HepG2 cells were infected with Mtb at a MOI of 10 and at the respective time points post-infection, the intracellular Mtb was isolated. In all experiments, the cells were treated with amikacin to remove the extracellular bacteria, followed by subsequent washing by 1X PBS. 4M Guanidine isocyanate was added to the flasks in equal amount to the media and the cells were scraped. The suspension was

centrifuged at 3500 rpm for 7 min and the pellet was resuspended in 350 µl of the RNA Lysis buffer of MN Kit (740955.250), followed by bead beating. RNA was isolated as per the manufacturer's protocol. Isolated RNA was treated with Dnase (PGM052, Puregene) for 1 hr to remove genomic DNA contamination. RNA integrity was checked by running it on a 2% agarose gel with RNA loading dye.

1 µg of RNA was reverse transcribed to cDNA using the Takara cDNA synthesis kit (6110A) as per the manufacturer's protocol. Gene expression analysis by quantitative real-time PCR was performed PowerUp SYBR Green PCR Thermo Fischer Scientific, (A25742) master mix in ABI 7500 FAST instrument. 16S rRNA was used as the normalizing control and comparative Ct method was used for quantification.

## Fluorescent fatty acid labeling of HepG2 and isolation

HepG2 cells were incubated with 7.5 µg/ml of fluorescently tagged fatty acid (BODIPY 558/568 $C_{12}$) for 24 hr. The unincorporated fatty acids were removed by washing it with 1X PBS, three times. After that, the cells were treated with DMSO or inhibitors (T863 and C75) for 24 hr, followed by infection with Mtb at a MOI of 10, following the standardized protocol of infection. A separate set of cells before infection were analysed for the effect of the inhibitors on LD formation by microscopy.

After, 48 hr post-infection, the cells were washed with 1X PBS and treated with Triton X –100 (0.05% v/v in water), probe sonicated, and Mtb from the cells were isolated by centrifuging at 3500 RPM for 8 min. Isolated Mtb cells were washed with 1X PBS and fixed with 4% PFA and stained with DAPI (1 mg/ml). The cells were mounted on poly-L-lysine-coated slides with an antifade agent. The cells were visualised in Zeiss 980 LSM confocal microscope at 100X, magnification.

## Gene expression studies

### RNA isolation

Total RNA was isolated from HepG2, PHCs, and liver sections using the MN-NucleoSpin RNA isolation kit (740955.250) following the manufacturer's protocol. For RNA sequencing, an equal number of cells was used during isolation. For liver tissue, approximately 10 mg was tissue was used. The quality of the RNA was verified by running it on a 1.5% agarose gel and by monitoring 260/280 ratios. All the RNA samples were frozen together in –80°C. For RNA sequencing, 3-5 µg of RNA was shipped in sodium acetate buffer (3 M Sodium acetate, pH 5.2) with 100% Ethanol. Four biological replicates from each time point were sent for sequencing. The RNA samples with RNA integrity number (RIN > 8.5) were used for library preparation.

RNA samples (for four biological replicates) were subjected to pair-end RNA sequencing after rRNA depletion on the Illumina platform Novaseq-6000 at CCMB, Hyderabad, India. Quality control and sequence trimming was performed using fastp (v0.23.2) (RRID:SCR_016962). The trimmed paired-end reads were aligned to the human genome (GRCh38) using the HISAT2 (v2.2.1) (RRID:SCR_015530) pipeline. Reads were assembled into transcripts using StringTie (v2.2.1) (RRID:SCR_016323). Annotation was conducted using aligned sequences and a GTF annotation file. The mapped reads were then used for generating the count table using StringTie (v2.2.1), genes lacking an Ensembl ID annotation were excluded. We arrived at a list of 62694 genes which were used for further analysis (available through GEO accession- GSE256184). The differential count was performed by DEseq2 (R package) using the uninfected samples of respective time points. Pathway enrichment was performed using the GO database and clusters were visualized using the R package ClusterProfiler. Further pathways of interest were analyzed by GAGE and visualized using Pathview (RRID:SCR_002732) in KEGG view.

### cDNA synthesis and RT-qPCR

1 µg of RNA was reverse transcribed to cDNA using the Takara cDNA synthesis kit (6110A) as per the manufacturer's protocol. Gene expression analysis by quantitative real-time PCR was performed PowerUp SYBR Green PCR Thermo Fischer Scientific, (A25742) master mix in ABI 7500 FAST instrument. Beta-actin was used as the normalizing control and comparative Ct method was used for quantification.

## List of primers used in the study

| GENE NAMES | FORWARD PRIMER (5'-------------->3') | REVERSE PRIMER (5'-------------->3') | |
|---|---|---|---|
| β-actin | ATGGAGGGGAATACAGCCC | TTCTTTGCAGCTCCTTCGTT | Mouse |
| ppar-γ | CTCTGGGAGATTCTCCTGTTGA | GGTGGGCCAGAATGGCATCT | Mouse |
| ppar-α | AGAGCCCCATCTGTCCTCTC | ACTGGTAGTCTGCAAAACCAAA | Mouse |
| fasn | CTGCGTGGCTATGATTATGG | AGGTTGCTGTCGTCTGTAGT | Mouse |
| acat1 | CAGGAAGTAAGATGCCTGGAAC | TTCACCCCCTTGGATGACATT | Mouse |
| acat2 | CCCGTGGTCATCGTCTCAG | GGACAGGGCACCATTGAAGG | Mouse |
| dgat1 | TGCCCTGACAGAGCAGATGG | CAGGTTGACATCCCGGTAGG | Mouse |
| dgat2 | TGGCGCTACTTCCGAGACTAC | TGCTGACTTCAGTAGCCTCTGTG | Mouse |
| nat2 | ACACTCCAGCCAATAAGTACAGC | GGTAGGAACGTCCAAACCCA | Mouse |
| Slco1b2 | GGGAACATGCTTCGTGGGATA | GGAGTTATGCGGACACTTCTC | Mouse |
| aadac | TACCGCTTCCAGATGCTATTGA | ACTGATTCCCAAAAGTTCACCAA | Mouse |
| cyp2e1 | CATCACCGTTGCCTTGCTTG | GGGGCAGGTTCCAACTTCT | Mouse |
| ces1c | GCACTACGCTGGGTCCAAGAT | AAAGATGGTCACGGAATCCG | Mouse |
| ces2c | GCTGAATGCTGGGTTCTTCG | GCTGCCTTGGATCTGTCCTGT | Mouse |
| β-actin | AAGGCCAACCGCGAGAAGAT | GCCAGAGGCGTACAGGGATA | Human |
| pparγ | GGTGAAACTCTGGGAGATTCT | CTCTGTGTCAACCATGGTCA | Human |
| fasn | AGTACACACCCAAGGCCAAG | GTGGATGATGCTGATGATGG | Human |
| acat1 | TACCAGAAGTAAAGCAGCATGG | TCATTCAGTGTACTGGCATTGG | Human |
| acat2 | CTTTAGCACGGATAGTTTCCTGG | GCTGCAAAGGCTTCATTGATTTC | Human |
| dgat1 | CAATCTGACCTACCGCGATCT | TCGATGATGCGTGAGTAGTCC | Human |
| dgat2 | GAATGGGAGTGGCAATGCTAT | CCTCGAAGATCACCTGCTTGT | Human |
| nat2 | CCAGAAGGGGTTTACTGTTTGG | CAGGTTTGGGCACGAGATTTC | Human |
| mgat1 | CGCAAGTTCCAGGGCTACTAC | CTTCAGCAGCGGATAGGTGG | Human |
| fsp27 | ATGGACTACGCCATGAAGTCT | CGGTGCTAACACGACAGGG | Human |
| Rv1258c | CTACGAGGCGATCCTCAACC | CGAGGATGGACAACCCGAAT | Mtb |
| Rv1819c | ATCGGGGTTTTCAGCGTGAT | GTCGAGCCAGTCTTGTGTGA | Mtb |
| Rv1218c | ATCGAGATTCGCGGACTGAC | ACATCGCCTGGAACATAGGC | Mtb |
| Rv1410c | ATGGTGACGCTGGTTGATGT | CGGTAGGGCGATAAGGAACC | Mtb |
| tgs1 (Rv3130c) | CCTTCTTATCGTCGCTCGCT | GCCAAGATCGAAGTCGGGAT | Mtb |
| tgs4 (Rv3088) | ACGGTCTCGTTCCTCAACAC | CCGGTGAATCGGAGAGGAAG | Mtb |
| lipY (Rv3097c) | TCTGCGCTCGAAACTCACTT | GCTCATCCCGTCATAGGTGG | Mtb |
| acrA1 (Rv3391) | ATGGCGGTCAACTACTTCGG | CCTTGGTGGGCAGATACGAG | Mtb |

## List of reagents in the study

| Reagent | Company | Catalogue number |
|---|---|---|
| Immobilon Forte Western HRP substrate | Millipore | WBLUF0500 |
| Amersham Hybond Western blotting membranes, PVDF | Merck | GE10600023 |
| PowerUP SYBR | Thermo Fischer Scientific | A25742 |

*Continued on next page*

*Continued*

| Reagent | Company | Catalogue number |
|---|---|---|
| Vectashield | Sigma-Aldrich | F6182 |
| Primescript 1st strand cDNA synthesis kit | Takara Bio | 6110 A |
| NucleoSpin RNA isolation Kit | Macherey-Nagel | 74106 |
| ProLong Gold antifade reagent | Thermo Fischer Scientific | P36930 |
| PolyFreeze Tissue Freezing medium | Sigma- Aldrich | SHH0026 |
| Microscope slides | Himedia | BG005 |
| Poly-Prep slides | Sigma- Aldrich | P0425-72EA |
| 2 X RNA loading dye | Thermo Scientific | R0641 |

## Softwares

All the graphs have been generated using GraphPad Prism 10 (RRID:SCR_002798, http://www.graphpad.com/) and Microsoft Excel (RRID:SCR_016137). The schematics have been drawn with the help of Biorender.com (RRID:SCR_018361).

## Acknowledgements

We thank Dr. Lakshyaveer Singh, Tuberculosis Aerosol Challenge Facility (TACF), ICGEB for mice experiments. We thank Dr. Neerja Wadhwa for helping in the NII Central Confocal facility, Mr. Birendra Nath Roy for the preparation of the cryosections, and the NII animal facility for providing us with the animals. We thank the Next Generation Sequencing (NGS) facility at CSIR-CCMB for transcriptomic support.

## Additional information

### Competing interests

Ashwani Kumar: Reviewing editor, *eLife*. The other authors declare that no competing interests exist.

### Funding

| Funder | Grant reference number | Author |
|---|---|---|
| Department of Biotechnology, Ministry of Science and Technology, India | | Rajesh S Gokhale |
| Science and Engineering Research Board | SB/SJF/2021-22/01 | Siddhesh S Kamat |
| Department of Science and Technology Fund for Improvement of S&T Infrastructure (DST-FIST) | SR/FST/LSII-043/2016 | Siddhesh S Kamat |

The funders had no role in study design, data collection and interpretation, or the decision to submit the work for publication.

### Author contributions

Binayak Sarkar, Conceptualization, Data curation, Formal analysis, Investigation, Visualization, Methodology, Writing – original draft, Writing – review and editing; Jyotsna Singh, Software, Formal analysis, Validation, Investigation, Methodology; Mohit Yadav, Shweta Singh, Aakash Chandramouli, Kritee Mehdiratta, Investigation, Methodology; Priya Sharma, Raman Deep Sharma, Formal analysis, Investigation, Methodology; Ashwani Kumar, Resources, Supervision, Validation, Investigation,

Methodology, Writing – original draft; Siddhesh S Kamat, Resources, Formal analysis, Supervision, Validation, Investigation, Visualization, Methodology, Writing – original draft; Devram S Ghorpade, Supervision, Validation, Investigation, Methodology; Debasisa Mohanty, Supervision, Funding acquisition, Validation, Investigation, Methodology, Writing – original draft, Project administration, Writing – review and editing; Dhiraj Kumar, Resources, Formal analysis, Supervision, Validation, Investigation, Methodology, Writing – original draft, Project administration; Rajesh S Gokhale, Conceptualization, Resources, Data curation, Formal analysis, Supervision, Funding acquisition, Validation, Investigation, Visualization, Methodology, Writing – original draft, Project administration, Writing – review and editing

### Author ORCIDs
Binayak Sarkar ⬤ https://orcid.org/0009-0009-3366-4851
Dhiraj Kumar ⬤ https://orcid.org/0000-0001-7578-2930
Rajesh S Gokhale ⬤ https://orcid.org/0000-0001-6597-2685

### Ethics

Paraffinized non-Mtb-infected, and Mtb-infected human autopsied liver specimens were obtained from the Postgraduate Institute of Medical Education and Research, Chandigarh, India. The Institutional Ethics Committee of CSIR-IMTech (Council of Scientific and Industrial Research-Institute of Microbial Technology) approved all research experiments involving human samples (Approval no [IEC (May 2021) #6]). The samples utilized in the study are from a library of autopsied human specimens in PGIMER (Postgraduate Institute of Medical Education and Research). The consent is provided by the relatives of the deceased person while its submission to the PGIMER. The details of the human subjects have been provided in the Supplementary file 1.

All the mice (C57BL/6) were housed in the animal house at the National Institute of Immunology (NII) before being transported to the TACF-ICGEB (Tuberculosis AerosolChallenge Facility), for subsequent animal infection studies. The animal experiments were performed adhering to the institutional guidelines (Approval number: Institutional Animal Ethical Committee, IAEC 543/20). Female Hartley Dunkin guinea pigs were used for infection studies in TACF-ICGEB in accordance with the institutional guidelines (Approval number: Institutional Animal Ethical Committee, IEAC #440/17).

Reviewer #1 (Public review): https://doi.org/10.7554/eLife.103817.3.sa1
Reviewer #2 (Public review): https://doi.org/10.7554/eLife.103817.3.sa2
Reviewer #3 (Public review): https://doi.org/10.7554/eLife.103817.3.sa3
Author response https://doi.org/10.7554/eLife.103817.3.sa4

## Additional files

### Supplementary files
MDAR checklist

Supplementary file 1. Details of the human biopsy samples used in the study.

Supplementary file 2. Differentially expressed genes between uninfected and Mtb-infected HepG2 at 0 hr and 48 hr post-infection.

### Data availability
All data generated or analyzed during this study are included in the manuscript and supporting files; source data files have been provided for all figures. RNA sequencing data were deposited in GEO with the accession number GSE256184.

The following dataset was generated:

| Author(s) | Year | Dataset title | Dataset URL | Database and Identifier |
| --- | --- | --- | --- | --- |
| Gokhale R, Mohanty D, Nandicoori VK, Sarkar B, Singh J | 2024 | Transciptomic changes in *Mycobacterium tuberculosis* (Mtb) infected hepatocytes at early and late time points of infection | https://www.ncbi.nlm.nih.gov/geo/query/acc.cgi?acc=GSE256184 | NCBI Gene Expression Omnibus, GSE256184 |

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
