## [Editor Report · eLife Assessment]

This **fundamental** study examines infection of the liver and hepatocytes during tuberculosis infection. The authors **convincingly** demonstrate that aerosol infection of mice and guinea pigs leads to appreciable infection of the liver as well as the lung. A further strength of the study lies in clinical evaluation of the presence of tuberculosis bacteria in human autopsied liver samples from individuals with miliary tuberculosis and the presence of a clear granuloma-like structure, which will prompt further study.

---

## [Referee Report · Reviewer #1 (Public review)]

Summary:

Authors showed the presence of Mtb in human liver biopsy samples of TB patient and reported that chronic infection of Mtb causes immune-metabolic dysregulation. Authors showed that Mtb replicates in hepatocytes in a lipid rich environment created by up regulating transcription factor PPARγ. Authors also reported that Mtb protects itself from anti-TB drugs by inducing drug metabolising enzymes.

Strengths:

It has been shown that Mtb induces storage of triacylglycerol in macrophages by induction of WNT6/ACC2 which helps in its replication and intracellular survival, however, creation of favorable replicative niche in hepatocytes by Mtb is not reported. It is known that Mtb infect macrophages and induces formation of lipid-laden foamy macrophages which eventually causes tissue destruction in TB patient. In a recent article it has been reported that "A terpene nucleoside from *M. tuberculosis* induces lysosomal lipid storage in foamy macrophages" that shows how Mtb manipulates host defense mechanisms for its survival. In this manuscript, authors reported the enhancement of lipid droplets in Mtb infected hepatocytes and convincingly showed that fatty acid synthesis and triacylglycerol formation is important for growth of Mtb in hepatocytes. Authors also showed the molecular mechanism for accumulation of lipid and showed that the transcription factor associated with lipid biogenesis, PPARγ and adipogenic genes were upregulated in Mtb infected cells.

The comparison of gene expression data between macrophages and hepatocytes by authors is important which indicates that Mtb modulates different pathways in different cell type as in macrophages it is related to immune response whereas, in hepatocytes it is related to metabolic pathways.

Authors also reported that Mtb residing in hepatocytes showed drug tolerance phenotype due to up regulation of enzymes involved in drug metabolism and showed that cytochrome P450 monooxygenase that metabolize rifampicin and NAT2 gene responsible for N-acetylation of isoniazid were up regulated in Mtb infected cells.

Weaknesses:

There are reports of hepatic tuberculosis in pulmonary TB patients especially in immune-compromised patients, therefore finding granuloma in human liver biopsy samples is not surprising.

Mtb infected hepatic cells showed induced DME and NAT and this could lead to enhanced metabolism of drug by hepatic cells as a result Mtb in side HepG2 cells get exposed to reduced drug concentration and show higher tolerance to drug. Authors mentioned that " hepatocyte resident Mtb may display higher tolerance to rifampicin". In my opinion higher tolerance to drug is possible only when DME of Mtb inside is up regulated or target is modified. Although, in the end authors mentioned that drug tolerance phenotype can be better attributed to host intrinsic factors rather than Mtb efflux pumps. It may be better if Drug tolerant phenotype section can be rewritten to clarify the facts.

In the revised manuscript, by immune-staining authors convincingly showed that hepatocytes are a favourable niche for replication of MTb.

Authors have rewritten the drug tolerant phenotype section which reads better.

Overall, this paper has new and important information on how MTb establishes a favourable niche for growth in hepatocytes and creates a drug tolerant environment.

---

## [Referee Report · Reviewer #2 (Public review)]

The manuscript by Sarkar et al has demonstrated the infection of liver cells/hepatocytes with Mtb and the significance of liver cells in the replication of Mtb by reprogramming lipid metabolism during tuberculosis. Besides, the present study shows that similar to Mtb infection of macrophages (reviewed in Chen et al., 2024; Toobian et al., 2021), Mtb infects liver cells but with a greater multiplication owing to consumption of enhanced lipid resources mediated by PPARg that could be cleared by its inhibitors. The strength of the study lies in clinical evaluation of the presence of Mtb in human autopsied liver samples from individuals with miliary tuberculosis and presence of a clear granuloma-like structure. The interesting observation is of granuloma-like structure in liver which prompts further investigations in the field.

The modulation of lipid synthesis during Mtb infection, such as PPARg upregulation, appears generic to different cell types including both liver cells and macrophage cells. It is also known that infection affect PPARγ expression and activity in hepatocytes. It is also known that this can lead to lipid droplet accumulation in the liver and the development of fatty liver disease (as shown for HCV). This study is in similar line for M.tb infection. As liver is the main site for lipid regulation, the availability of lipid resources is greater and higher is the replication rate. In short, the observations from the study confirm the earlier studies with these additional cell types. It is known that higher the lipid content, greater are Lipid Droplet-positive Mtb and higher is the drug resistance (Mekonnen et al., 2021). The DMEs of liver cells add further to the phenotype.

Comments on revised version:

The authors noted that even in experiments where mice were infected with lower CFUs, the presence of Mtb colonies could still be detected in the liver. It would be beneficial to include some experimental data related to this in the supplementary information, as it could provide valuable insights for the research field.

---

## [Referee Report · Reviewer #3 (Public review)]

In this revised manuscript, the authors explore how Mtb can infect hepatocytes and create a favorable niche associated with upregulation of the transcription factor PPARγ which presumably allows the bacteria to scavenge lipids from lipid droplets in host cells and upregulate drug-metabolizing enzymes to protect against its elimination. In response to the review, the authors have performed some additional immunostaining of hepatocytes, added more detail to figure legends, added experiments somewhat showing improved colocalization and staining, clarified several points and paragraphs, and updated the referenced literature and discussion.

The current manuscript provides evidence that human miliary TB patients have infection of hepatocytes with Mtb, with evidence that the bacteria survive at least partially through upregulation of PPARγ, which significantly changes the lipid milieu of the cells. There is also an examination of transcriptomics and lipid metabolism in response to Mtb infection, as well as drug tolerance of Mtb inside hepatocytes. The current manuscript is an improvement over the previous one.

However, although the manuscript is improved, tissue immunophenotyping of the various cells in the liver remains weak and unconvincing. This is truly a missed opportunity and lessens the rigor of the central findings and conclusions. As pointed out by another reviewer, literature has described different fates of Mtb in the liver. Given the tissue available to the authors, carefully dissecting the various cells that the bacteria are in (esp. hepatocytes versus Kupffer cells) is critical. The authors use only 2 generic markers and do not distinguish among cell types within the tissue slices. A review of the literature shows a variety of both human and mouse antibody markers. In fact, a liver atlas based on immunophenotyping has been published. Likewise, the authors comment on liver granulomas, but this is not justified without immunophenotyping.

---

## [Author Response]

The following is the authors’ response to the current reviews.

**Public Reviews:**

**Reviewer #1 (Public review):**
Summary:Authors showed the presence of Mtb in human liver biopsy samples of TB patient and reported that chronic infection of Mtb causes immune-metabolic dysregulation. Authors showed that Mtb replicates in hepatocytes in a lipid rich environment created by up regulating transcription factor PPARγ. Authors also reported that Mtb protects itself from anti-TB drugs by inducing drug metabolising enzymes.Strengths:It has been shown that Mtb induces storage of triacylglycerol in macrophages by induction of WNT6/ACC2 which helps in its replication and intracellular survival, however, creation of favorable replicative niche in hepatocytes by Mtb is not reported. It is known that Mtb infect macrophages and induces formation of lipid-laden foamy macrophages which eventually causes tissue destruction in TB patient. In a recent article it has been reported that "A terpene nucleoside from *M. tuberculosis* induces lysosomal lipid storage in foamy macrophages" that shows how Mtb manipulates host defense mechanisms for its survival. In this manuscript, authors reported the enhancement of lipid droplets in Mtb infected hepatocytes and convincingly showed that fatty acid synthesis and triacylglycerol formation is important for growth of Mtb in hepatocytes. Authors also showed the molecular mechanism for accumulation of lipid and showed that the transcription factor associated with lipid biogenesis, PPARγ and adipogenic genes were upregulated in Mtb infected cells.The comparison of gene expression data between macrophages and hepatocytes by authors is important which indicates that Mtb modulates different pathways in different cell type as in macrophages it is related to immune response whereas, in hepatocytes it is related to metabolic pathways.Authors also reported that Mtb residing in hepatocytes showed drug tolerance phenotype due to up regulation of enzymes involved in drug metabolism and showed that cytochrome P450 monooxygenase that metabolize rifampicin and NAT2 gene responsible for N-acetylation of isoniazid were up regulated in Mtb infected cells.Weaknesses:There are reports of hepatic tuberculosis in pulmonary TB patients especially in immune-compromised patients, therefore finding granuloma in human liver biopsy samples is not surprising.Mtb infected hepatic cells showed induced DME and NAT and this could lead to enhanced metabolism of drug by hepatic cells as a result Mtb in side HepG2 cells get exposed to reduced drug concentration and show higher tolerance to drug. Authors mentioned that " hepatocyte resident Mtb may display higher tolerance to rifampicin". In my opinion higher tolerance to drug is possible only when DME of Mtb inside is up regulated or target is modified. Although, in the end authors mentioned that drug tolerance phenotype can be better attributed to host intrinsic factors rather than Mtb efflux pumps. It may be better if Drug tolerant phenotype section can be rewritten to clarify the facts.In the revised manuscript, by immune-staining authors convincingly showed that hepatocytes are a favourable niche for replication of MTb.Authors have rewritten the drug tolerant phenotype section which reads better.Overall, this paper has new and important information on how MTb establishes a favourable niche for growth in hepatocytes and creates a drug tolerant environment.

We thank the reviewer for the through and insightful review.

**Reviewer #2 (Public review)**:The manuscript by Sarkar et al has demonstrated the infection of liver cells/hepatocytes with Mtb and the significance of liver cells in the replication of Mtb by reprogramming lipid metabolism during tuberculosis. Besides, the present study shows that similar to Mtb infection of macrophages (reviewed in Chen et al., 2024; Toobian et al., 2021), Mtb infects liver cells but with a greater multiplication owing to consumption of enhanced lipid resources mediated by PPARg that could be cleared by its inhibitors. The strength of the study lies in clinical evaluation of the presence of Mtb in human autopsied liver samples from individuals with miliary tuberculosis and presence of a clear granuloma-like structure. The interesting observation is of granuloma-like structure in liver which prompts further investigations in the field.The modulation of lipid synthesis during Mtb infection, such as PPARg upregulation, appears generic to different cell types including both liver cells and macrophage cells. It is also known that infection affect PPARγ expression and activity in hepatocytes. It is also known that this can lead to lipid droplet accumulation in the liver and the development of fatty liver disease (as shown for HCV). This study is in similar line for M.tb infection. As liver is the main site for lipid regulation, the availability of lipid resources is greater and higher is the replication rate. In short, the observations from the study confirm the earlier studies with these additional cell types. It is known that higher the lipid content, greater are Lipid Droplet-positive Mtb and higher is the drug resistance (Mekonnen et al., 2021). The DMEs of liver cells add further to the phenotype.Comments on revised version:The authors noted that even in experiments where mice were infected with lower CFUs, the presence of Mtb colonies could still be detected in the liver. It would be beneficial to include some experimental data related to this in the supplementary information, as it could provide valuable insights for the research field.

We thank the reviewer for the in depth evaluation of our manuscript and as suggested we will include the data where Mtb was detected in the liver at low CFUs

**Reviewer #3 (Public review):**
In this revised manuscript, the authors explore how Mtb can infect hepatocytes and create a favorable niche associated with upregulation of the transcription factor PPARγ which presumably allows the bacteria to scavenge lipids from lipid droplets in host cells and upregulate drug-metabolizing enzymes to protect against its elimination. In response to the review, the authors have performed some additional immunostaining of hepatocytes, added more detail to figure legends, added experiments somewhat showing improved colocalization and staining, clarified several points and paragraphs, and updated the referenced literature and discussion.The current manuscript provides evidence that human miliary TB patients have infection of hepatocytes with Mtb, with evidence that the bacteria survive at least partially through upregulation of PPARγ, which significantly changes the lipid milieu of the cells. There is also an examination of transcriptomics and lipid metabolism in response to Mtb infection, as well as drug tolerance of Mtb inside hepatocytes. The current manuscript is an improvement over the previous one.However, although the manuscript is improved, tissue immunophenotyping of the various cells in the liver remains weak and unconvincing. This is truly a missed opportunity and lessens the rigor of the central findings and conclusions. As pointed out by another reviewer, literature has described different fates of Mtb in the liver. Given the tissue available to the authors, carefully dissecting the various cells that the bacteria are in (esp. hepatocytes versus Kupffer cells) is critical. The authors use only 2 generic markers and do not distinguish among cell types within the tissue slices. A review of the literature shows a variety of both human and mouse antibody markers. In fact, a liver atlas based on immunophenotyping has been published. Likewise, the authors comment on liver granulomas, but this is not justified without immunophenotyping.

We would like to thank the reviewer for the in-depth and detailed suggestions. We would like to clarify that the primary aim of our study was to determine the localization of Mtb within hepatocytes and the downstream biological consequences. To this end, we employed two well-established and widely validated markers (ASPGR 1 and albumin) that are consistently used to identify hepatocytes in both human and murine liver tissue. While we acknowledge the broader potential of comprehensive immunophenotyping, our focused approach was designed to specifically address the question of hepatocyte involvement, which the selected markers effectively support, which was further reiterated by the Reviewer 1.

**Recommendations for the authors:**

**Reviewer #1 (Recommendations for the authors):**
In my opinion this paper contains important information and no further information is required for this manuscript.

We thank the reviewer for the insightful comments

**Reviewer #2 (Recommendations for the authors):**
The authors noted that even in experiments where mice were infected with lower CFUs, the presence of Mtb colonies could still be detected in the liver. It would be beneficial to include some experimental data related to this in the supplementary information, as it could provide valuable insights for the research field.

As suggested, we will include the data with the low CFUs in the updated manuscript.

**Reviewer #3 (Recommendations for the authors):**
• Line 340, the fact that PPARγ inhibition decreases bacterial load should not be surprising, as the authors cite several papers where this is already shown.• Line 379, the increased tolerance of Mtb to drugs in hepatocytes is only significant at the lower 2 concentrations, not at 5 ug/mL.• Fig S4F-H, the y axis is inappropriately not set to zero on the lower limit.• Fig S9B, the Y-axis states "relative" CFU, but there is no indication what the bars are normalized to, and the numbers are much more typical of standard CFU values. Was the "Relative" part left in by mistake?• Double check the ending of the figure legend for Figure S10 and S11.• Line 352, phenomenom [sic] is misspelled.• On re-read, several sentences throughout this manuscript need improvement regarding structure and grammar. I suggest careful editorial review.

We thank the reviewer for pointing out the issues and these will be carefully modified in the next version.

The following is the authors’ response to the original reviews

**Public Reviews:**

**Reviewer #1 (Public review):**
Summary:The authors showed the presence of Mtb in human liver biopsy samples of TB patients and reported that chronic infection of Mtb causes immune-metabolic dysregulation. Authors showed that Mtb replicates in hepatocytes in a lipid rich environment created by up regulating transcription factor PPARγ. Authors also reported that Mtb protects itself from anti-TB drugs by inducing drug metabolising enzymes.Strengths:It has been shown that Mtb induces storage of triacylglycerol in macrophages by induction of WNT6/ACC2 which helps in its replication and intracellular survival, however, creation of favorable replicative niche in hepatocytes by Mtb is not reported. It is known that Mtb infects macrophages and induces formation of lipid-laden foamy macrophages which eventually causes tissue destruction in TB patients. In a recent article it has been reported that "A terpene nucleoside from *M. tuberculosis* induces lysosomal lipid storage in foamy macrophages" that shows how Mtb manipulates host defense mechanisms for its survival. In this manuscript, authors reported the enhancement of lipid droplets in Mtb infected hepatocytes and convincingly showed that fatty acid synthesis and triacylglycerol formation is important for growth of Mtb in hepatocytes. The authors also showed the molecular mechanism for accumulation of lipid and showed that the transcription factor associated with lipid biogenesis, PPARγ and adipogenic genes were upregulated in Mtb infected cells.The comparison of gene expression data between macrophages and hepatocytes by authors is important which indicates that Mtb modulates different pathways in different cell type as in macrophages it is related to immune response whereas, in hepatocytes it is related to metabolic pathways.Authors also reported that Mtb residing in hepatocytes showed drug tolerance phenotype due to up regulation of enzymes involved in drug metabolism and showed that cytochrome P450 monooxygenase that metabolize rifampicin and NAT2 gene responsible for N-acetylation of isoniazid were up regulated in Mtb infected cells.

We thank the reviewer for the positive feedback and for highlighting the strengths of our study.

Weaknesses:There are reports of hepatic tuberculosis in pulmonary TB patients especially in immune-compromised patients, therefore finding granuloma in human liver biopsy samples is not surprising.Mtb infected hepatic cells showed induced DME and NAT and this could lead to enhanced metabolism of drug by hepatic cells as a result Mtb in side HepG2 cells get exposed to reduced drug concentration and show higher tolerance to drug. The authors mentioned that " hepatocyte resident Mtb may display higher tolerance to rifampicin". In my opinion higher tolerance to drugs is possible only when DME of Mtb inside is up regulated or the target is modified. Although, in the end authors mentioned that drug tolerance phenotype can be better attributed to host intrinsic factors rather than Mtb efflux pumps. It may be better if the Drug tolerant phenotype section can be rewritten to clarify the facts.

We agree that several case studies regarding liver infection in pulmonary TB patients have been reported in the literature, however this report is the first comprehensive study that establishes hepatocytes to be a favourable niche for Mtb survival and growth.

Drug tolerance is a phenomenon that is exhibited by the bacteria and during hostpathogen interactions, can be influenced by both intrinsic (bacterial) and extrinsic (host-mediated) factors. Multiple examples of tolerance being attributed to host driven factors can be found in literature (PMID 32546788, PMID: 28659799, PMID: 32846197). Our studies demonstrate that Mtb infected hepatocytes create a drug tolerant environment by modulating the expression of Drug modifying enzymes (DMEs) in the hepatocytes.

As suggested by the reviewer we will rewrite the drug tolerant phenotype section.

**Reviewer #2 (Public review):**
The manuscript by Sarkar et al has demonstrated the infection of liver cells/hepatocytes with Mtb and the significance of liver cells in the replication of Mtb by reprogramming lipid metabolism during tuberculosis. Besides, the present study shows that similar to Mtb infection of macrophages (reviewed in Chen et al., 2024; Toobian et al., 2021), Mtb infects liver cells but with a greater multiplication owing to consumption of enhanced lipid resources mediated by PPARg that could be cleared by its inhibitors. The strength of the study lies in the clinical evaluation of the presence of Mtb in human autopsied liver samples from individuals with miliary tuberculosis and the presence of a clear granuloma-like structure. The interesting observation is of granuloma-like structure in liver which prompts further investigations in the field.The modulation of lipid synthesis during Mtb infection, such as PPARg upregulation, appears generic to different cell types including both liver cells and macrophage cells. It is also known that infection affect PPARγ expression and activity in hepatocytes. It is also known that this can lead to lipid droplet accumulation in the liver and the development of fatty liver disease (as shown for HCV). This study is in a similar line for M.tb infection. As the liver is the main site for lipid regulation, the availability of lipid resources is greater and higher is the replication rate. In short, the observations from the study confirm the earlier studies with these additional cell types. It is known that higher the lipid content, the greater are Lipid Droplet-positive Mtb and higher is the drug resistance (Mekonnen et al., 2021). The DMEs of liver cells add further to the phenotype.

We thank the reviewer for emphasizing on the strengths of our study and how it can lead to further investigations in the field.

**Reviewer #3 (Public review):**
This manuscript by Sarkar et al. examines the infection of the liver and hepatocytes during *M. tuberculosis* infection. They demonstrate that aerosol infection of mice and guinea pigs leads to appreciable infection of the liver as well as the lung. Transcriptomic analysis of HepG2 cells showed differential regulation of metabolic pathways including fatty acid metabolic processing. Hepatocyte infection is assisted by fatty acid synthesis in the liver and inhibiting this caused reduced Mtb growth. The nuclear receptor PPARg was upregulated by Mtb infection and inhibition or agonism of its activity caused a reduction or increase in Mtb growth, respectively, supporting data published elsewhere about the role of PPARg in lung macrophage Mtb infection. Finally, the authors show that Mtb infection of hepatocytes can cause upregulation of enzymes that metabolize antibiotics, resulting in increased tolerance of these drugs by Mtb in the liver.Overall, this is an interesting paper on an area of TB research where we lack understanding. However, some additions to the experiments and figures are needed to improve the rigor of the paper and further support the findings. Most importantly, although the authors show that Mtb can infect hepatocytes in vitro, they fail to describe how bacteria get from the lungs to the liver in an aerosolized infection. They also claim that "PPARg activation resulting in lipid droplets formation by Mtb might be a mechanism of prolonging survival within hepatocytes" but do not show a direct interaction between PPARg activation and lipid droplet formation and lipid metabolism, only that PPARg promotes Mtb growth. Thus, the correlations with PPARg appear to be there but causation, implied in the abstract and discussion, is not proven.The human photomicrographs are important and overall, well done (lung and liver from the same individuals is excellent). However, in lines 120-121, the authors comment on the absence of studies on the precise involvement of different cells in the liver. In this study there is no attempt to immunophenotype the nature of the cells harboring Mtb in these samples (esp. hepatocytes). Proving that hepatocytes specifically harbor the bacteria in these human samples would add significant rigor to the conclusions made.

We thank the reviewer for nicely summarizing our manuscript.

Our study establishes the involvement of liver and hepatocytes in pulmonary TB infection in mice. Understanding the mechanism of bacterial dissemination from the lung to the liver in aerosol infections demands a detailed separate study.

Figure 6E and 6F shows how PPARγ agonist and antagonist modulate (increase and decrease respectively) bacterial growth in hepatocytes (further supported by the CFU data in Supplementary Figure 9B). Again, the number of lipid droplets in hepatocytes increase and decrease with the treatment of PPARγ agonist and antagonist respectively as shown in Figure 6G and 6H. Collectively, these studies provide strong evidence that PPARγ activation leads to more lipid droplets that support better Mtb growth.

We thank the reviewer for finding our human photomicrographs convincing. In the manuscript, we provide evidence for the direct involvement of the hepatocytes (and liver) in Mtb infection. We have performed detailed immunophenotyping of hepatocyte cells in the mice model with ASPGR1 (asialoglycoprotein receptor 1) and in the revised version of record, we have further stained the infected hepatocytes with anti-albumin antibody.

**Recommendations for the authors:**

**Reviewer #1 (Recommendations for the authors):**
In my opinion drug tolerant phenotype section should be rewritten for better clarification. The manuscript contains important information about hepatic tuberculosis which are not reported yet.

We have rewritten the drug tolerant phenotype section for better clarity.

We appreciate the reviewer’s comments regarding important information about hepatic tuberculosis

**Reviewer #2 (Recommendations for the authors):**
The following are some observations and comments on the manuscript.(1) The study delves into the mechanisms related to hepatic TB/miliary TB; however, the introduction and discussion only describe and discuss the data in the context of pulmonary TB giving a sense that the mandate of the MS is the exploration of the role of liver cells in pulmonary TB. There appears a gap in the connection of findings from the Miliary TB to the pulmonary TB. A discussion of the conversion of pulmonary TB to extrapulmonary /hepatic TB in the light of the findings may be helpful.

We have modified the discussion section to include possible mechanisms that convert pulmonary TB to hepatic TB in the light of findings. Briefly, Pulmonary tuberculosis (TB) can lead to miliary TB probably through hematogenous dissemination, where Mtb spreads from the infected lungs into blood vessels either from a primary lung focus, reactivated TB or caseous necrosis. Once in blood vessels, the bacteria seed multiple organs, forming tiny granulomas, characteristic of miliary TB. The liver involvement could be either through direct hematogenous spread or extrusion from nearby infected lymph nodes, leading to hepatic TB, which presents with granulomas and liver dysfunction. This spread underscores the severity of untreated pulmonary TB and the need for early intervention. Our in vivo infection data clearly shows that pulmonary infection of Mtb in mice and guinea pigs can steadily leads to significant infection of the liver and metabolic abnormalities in the liver. The study further highlights the need for systemic studies to better understand the route and mode of dissemination from lungs to liver for better pathophysiological understanding of the disease and creating new therapeutic targets.

(2) The authors show the presence of Mtb in the liver autopsies of miliary tuberculosis patients. It is well known that Mtb disseminates during the late stages to several organs and liver is a major site (Sharma et al. 2005; 10.1016/S1473-3099(05)70163-8). Other clinical observations also point to the fact that although Mtb infects liver cells, it is cleared (Thandi et al., 2018, https://doi.org/10.4049/jimmunol.200.Supp.173.20). As the samples are from miliary TB, it is expected that the bacterial load must have been very high before spreading to blood. It is known that once in blood, M.tb is expected to spread to various organs, especially highly vascular ones. Were any other tissues (especially with high vasculature) stained and verified? If yes, add to the supplementary data or discuss.

Other tissues were not collected and stained during this study. Studies are currently underway to understand whether other vasculated organs also harbour Mtb or not. Besides several studies have shown that Mtb can infect a wide range of organs like brain, kidney, bone marrow, etc (PMID: 33142108, PMID: 28046053, PMID: 34269789) during miliary conditions.

(3) It is not evident from this paper if hepatic infiltration occurs in pulmonary TB patients? It may therefore be important to discuss the status of liver infections in the primary pulmonary infection.

Based on the available data from human biopsied liver samples, there is an indication of liver involvement in systemic tuberculosis (TB). However, to gain a more comprehensive understanding of hepatic infiltration in pulmonary TB patients, it is essential to conduct well-organized clinical studies. These studies should specifically target pulmonary TB patients and explore the extent and nature of liver involvement in these individuals (discussion). As suggested by the reviewer it is in the discussion

(4) Similarly, in the mice model, M.tb was shown to localize to liver when aerosolic infection was given. Were any other tissues, such as kidney, bone marrow etc, checked? Is it because of the high dose of M.tb against the standard challenge dose of 50-100 CFU? Further, since the study in the mouse model is to mimic a miliary tuberculosis of liver, did the dissemination occur via bloodstream and if mycobacteremia could be observed in infected mice.

Currently studies are underway to understand the involvement of other organs like kidney, brain, bone marrow, in aerosol infection mice model and how dissemination occurs in those distant organs.

The focus of the current study was to understand the role of liver in systemic tuberculosis with emphasis on hepatocytes as a key cell type to be infected. We have also conducted the experiments with lower CFUs and could detect the presence of Mtb colonies in liver, so we do not think that the infection of liver is dependent on the dose of infection.

(5) There are studies in mouse model which infer that liver carried the lowest bacterial burden, was cleared the fastest, and it is established that as compared to sites persistently seeded by *M. tuberculosis*, in the liver the bacteria rarely infect cell types other than professional phagocytes. As the observations in this study are contrasting, the discussion section should include a critical comparative analysis to justify why in the conditions used in the study, the hepatocytes and not Kupffer cells are infected. Other than the morphological description to indicate M.tb infection of hepatocytes in the liver section (fig 1E), it will be good to show localization of M.tb specifically to hepatocytes by using hepatocyte specific marker. Unlike as reported, why was a clearance of M.tb not observed even after 10 weeks (figure 2B).

While some studies show that Mtb from the liver is cleared fast but there are several other studies that report Liver harbours Mtb even after 10 weeks postinfection (PMID: 22359543, PMID: 21533158, PMID: 29242198). We have consistently observed Mtb infection of liver post week 10 in our infection model.

We have performed detailed immunophenotyping of hepatocyte cells in the mice model with ASPGR1 (asialoglycoprotein receptor 1) and in the revised version of record, we have further stained the isolated hepatocytes with anti-albumin antibody (albumin is a robust marker of hepatocyte identity) and have showed the presence of Mtb in it. The data has been included in the revised manuscript (Fig 2J)

(6) While the result section mentions that "individuals with miliary tuberculosis' (line 107), the legend of Figure 1 writes 'Presence of Mtb in human pulmonary tuberculosis patients'. This is confusing. Clarify

We thank the reviewer for pointing it out, we have changed the figure legends to miliary tuberculosis as most of the liver biopsy samples were obtained from military tuberculosis patients.

(7) Supplementary Figure 2D: Corresponding control panel (uninfected) should be added, which will also verify the specificity of Ag85b. As it is known that Ag85B is secreted out from the bacteria and hence the detected signals may not confirm that Mtb is in hepatocytes. Ag85B per bacterium decreases by almost 10,000-fold at later stages of infection because of secretion (Ernst JD, Cornelius A, et al 2019 mBio). In Supl figure 2D, Ag85b signal seems to be present everywhere inside the cells. Hence, it is important that the control panel be added.

We have included a control image below which shows no staining of Ag85B in the uninfected sample.While we acknowledge with the reviewer’s comment, but Ag85B has been consistently used as a marker for Mtb presence in multiple studies. Nargan et al., uses Ag85B based staining to characterize infection both pulmonary and EPTB samples (PMID: 38880068). Jain et al., uses Ag85B to characterize Mtb infection of Mesenchymal stem cell in lung biopsy samples of pulmonary TB patients (PMID: 32546788)

**Author response image 1. sa4fig1:** Ag85B staining in uninfected mice shows no signals.

(8) The kinetics experiments in Figure 3D-3G should have used time laps microscopy of a few of the infected cells or it should be represented in CFU. If we consider the doubling time of H37Rv is about 22h to 24h, the data showing that MFI increases dramatically from 5 HPI to 120 HPI, gives an impression that the bacterial number inside the cells increased more than its doubling time.

We have added the modified plot. As suggested, the CFU of Mtb within HepG2, PHCs, THP-1, RAW 264.7 and BMDMs have been included in the revised version (Supplementary Figure 4 D-H)

(9) What is the effect of C45 and T863 on Mtb growth invitro? The effect of C45 and T863 on Mtb growth invitro should be shown to be ruled out. The representative image in Figure 5F is DMSO or C45 treated cells panel? Please specify it.

As per the reviewer’s suggestion we have seen the effect of C45 (30 µM) and T863 (25 µM) on Mtb growth in vitro and did not find any difference in the growth kinetics. The representative image in Figure 5F is DMSO treated cells.

**Author response image 2. sa4fig2:** Growth kinetics of Mtb in 7H9 medium with DMSO, C75 and T863.

(10) Supplementary Figure 6B: Correct the Y-axis label from mRNA levels to Fold change (normalised to control). Please do similar changes wherever required.

We have made the necessary changes as per the suggestion of the reviewer.

(11) Figure 7B and 7C: How was the normalization performed? Is the data normalized to the number of bacteria that entered the specific cell type or was normalized at 48hrs with respect to DMSO? DMSO alone data should be shown.

In the drug tolerance assays, we have calculated the ratio of the bacterial burden in hepatocytes treated with drugs compared to hepatocytes treated with DMSO. The infection was given for 48 hours post which the infected cells were treated with the mentioned concentrations of isoniazid and rifampicin for 24 hours. CFU enumeration was conducted after this 24 hour. Figure 7A gives a schematic of the experimental set up.

% Tolerant Bacterial population = [A/B X 100] % where A is the CFU of Mtb from infected hepatocytes treated with drug and B is the CFU of Mtb infected cells treated with DMSO.Thus the effect of MOI is negated.

To provide further credence to the CFU data, we have analysed these studies using microscopic studies as well, where no cell death was observed under the conditions. Mouse BMDMs were as a macrophage control. We have calculated the % tolerance as ratio by measuring the mean fluorescent intensity of GFP-Mtb per hepatocyte treated with drug to MFI of GFP-Mtb per hepatocyte treated with DMSO (control). More than 20 fields, each consisting of more than 4 infected cells have been used for analysis providing additional evidence of less killing of Mtb in hepatocytes compared to BMDMs with anti-TB drugs. All these details are included in the manuscript.

(12) While authors have shown the changes in mRNA levels of CYP3A4, CYP3A43, NAT2, the protein or activities of some of these should be measured to verify the effect.

Currently studies are underway to understand the activities of the key proteins involved in isoniazid and rifampicin metabolism and will be published as a separate manuscript.

**Reviewer #3 (Recommendations for the authors):**
Additional comments are:• Figure 2D, the 20X and 40X magnifications do not look appreciably different in size. Please double-check that the correct images were used.

We thank the reviewer for pointing it out, we havecorrected it in the revised version.

• Lines 162-164: The authors state almost 100% purity. However, the contour plot in 2F appears to show 2 cell populations. Figure 2G is missing a legend of which colors correspond to which staining (and again there appears to be highly variable staining).

We agree with the reviewer that there are two contours observed in Figure 2F. Although both the contours are positive for ASPGR1 protein, but the level of expression of the ASPGR1 protein is variable. The corresponding confocal image (Nucleus stained by DAPI and ASPGR1 stained with ASPGR1 antibody with Alexa fluor 555 conjugated secondary antibody) also indicates a variable staining of isolated primary hepatocytes, where some cells give a stronger intensity signal than the other cells, further visually confirming our statement. Moreover, several studies show differential expression of ASPGR1 protein in hepatocyte like cells (PMID: 27143754)

To further clarify and be more specific with respect to the identity of the hepatocytes, we have stained primary hepatocytes from infected mouse livers with Albumin antibody (a stable marker for hepatocytes) and Ag85B (2J)

Multiple figures throughout the manuscript, including this one, would benefit from the use of arrows to depict what is described in the legend and text more clearly, and the use of higher power insets to better define cell architecture. Finally, some images appear blurry to the eye. Improvements are needed throughout.

As per the suggestion, we have modified the figures and figure legends for better clarity.

• Lines 153-155. Albumin, AST and GGT appear to be significantly up at week 8, contradicting the statement that there is no change until week 10.

We thank the reviewer for poiting it out and have made suitable changes in the write up

• Lines 203-205: The authors state earlier that bacteria survive in macrophage phagosomes. Do the authors know the niche for bacteria in hepatocytes that enable them to continue to grow? Transcriptome data from HepG2 cells suggest perhaps a phagosomal pathway?

We thank the reviewer for this insightful question. As rightly pointed out by the reviewer, transcription data indeed suggests changes in several important pathways like macroautophagy, golgi vesicular transport and vacuolar transport, which can affect the subcellular localisation of Mtb within hepatocytes. High resolution microscopic studies with respect to the subcellular localisation of labelled Mtb within Primary hepatocytes, HepG2 and THP-1 has been conducted and the % colocalization within different intra-cellular compartments have been measured. The image of colocalization of labelled Mtb within PHCs is shown below along with the % colocalization within various compartments in PHCs, HepG2 and THP-1 is added.

**Author response image 3. sa4fig3:** Colocalisation of Mtb-GFP with various intra-cellular markers within PHCs.

**Author response image 4. sa4fig4:** Percentage Colocalisation of Mtb-GFP with various intra-cellular markers within PHCs, HepG2 and THP-1.

• Validation of some critical genes found in the HepG2 cells should be done by qRTPCR in primary hepatocytes.

qRT-PCR analysis of some of the key genes in HepG2 have been validated in primary hepatocytes at 24 hours post infection. Majority of the genes show a similar trend.

**Author response image 5. sa4fig5:** Gene expression analysis of the mentioned genes in Mtb infected PHCs as compared to the uninfected control.

• Lines 259-260: The authors state a high degree of co-localization. The photomicrograph of a single cell in Fig. 5D is not convincing. I'm not even sure that they are really in the same subcellular compartment. Co-localization stated in Fig. S8B is also not convincing as shown.

The image currently shown in figure 3D is a maximum intensity projection image of multiple z-stacks encompassing the entire cell.

We agree with the reviewer with respect to figure Fig S8B and will modify the text and the figure legend accordingly.

Copywriting edits:• It is difficult to see individual gene names in Figures 4D and 4E. A higher resolution or larger font would be appreciated for the reader.

An excel file with the top differentially regulated genes at both 0 hours post infection and 48 hours post infection has been added.

• Figure 5A has a shadow on the top right image.

We have changed the image in the revised manuscript

• Figure 5E is difficult to read the labels on the axes; it would be better in general to make the labels separately instead of relying on the graphing software, since these labels can get stretched when the size of the graph is modified.

We agree with the reviewer and have made necessary changes.

• Line 163: should be "percent" and not "perfect."

We thank the reviewer for pointing it out and have corrected it

• Line 190: is missing a period at the end of the sentence "...for further experiments"

We thank the reviewer for pointing it out and have corrected it

• Line 332: should be "hepatocytes" instead of "hepatoctyte" [sic]

We thank the reviewer for pointing it out and have corrected it